# Offline congestion games: How feedback type affects data coverage requirement

**Haozhe Jiang**[1]* **Qiwen Cui**[2]* **Zhihan Xiong**[2] **Maryam Fazel**[2] **Simon S. Du**[2]

[1] Institute for Interdisciplinary Information Sciences, Tsinghua University
[2] University of Washington

## Abstract

This paper investigates when one can efficiently recover an approximate Nash Equilibrium (NE) in offline congestion games. The existing dataset coverage assumption in offline general-sum games inevitably incurs a dependency on the number of actions, which can be exponentially large in congestion games. We consider three different types of feedback with decreasing revealed information. Starting from the facility-level (a.k.a., semi-bandit) feedback, we propose a novel one-unit deviation coverage condition and show a pessimism-type algorithm that can recover an approximate NE. For the agent-level (a.k.a., bandit) feedback setting, interestingly, we show the one-unit deviation coverage condition is not sufficient. On the other hand, we convert the game to multi-agent linear bandits and show that with a generalized data coverage assumption in offline linear bandits, we can efficiently recover the approximate NE. Lastly, we consider a novel type of feedback, the game-level feedback where only the total reward from all agents is revealed. Again, we show the coverage assumption for the agent-level feedback setting is insufficient in the game-level feedback setting, and with a stronger version of the data coverage assumption for linear bandits, we can recover an approximate NE. Together, our results constitute the first study of offline congestion games and imply formal separations between different types of feedback.

## 1 Introduction

Congestion game is a special class of general-sum matrix games that models the interaction of players with shared facilities (Rosenthal, 1973). Each player chooses some facilities to utilize, and each facility will incur a different reward depending on how congested it is. For instance, in the routing game (Koutsoupias & Papadimitriou, 1999), each player decides a path to travel from the starting point to the destination point in a traffic graph. The facilities are the edges and the joint decision of all the players determines the congestion in the graph. The more players utilize one edge, the longer the travel time on that edge will be. As one of the most well-known classes of games, congestion game has been successfully deployed in numerous real-world applications such as resource allocation (Johari & Tsitsiklis, 2003), electrical grids (Ibars et al., 2010) and cryptocurrency ecosystem (Altman et al., 2019).

Nash equilibrium (NE), one of the most important concepts in game theory (Nash Jr, 1950), characterizes the emerging behavior in a multi-agent system with selfish players. It is commonly known that solving for the NE is computationally efficient in congestion games as they are isomorphic to potential games (Monderer & Shapley, 1996). Assuming full information access, classic dynamics such as best response dynamics (Fanelli et al., 2008), replicator dynamics (Drighes et al., 2014) and no-regret dynamics (Kleinberg et al., 2009) provably converge to NE in congestion games. Recently Heliou et al. (2017) and Cui et al. (2022) relaxed the full information setting to the online (semi-)bandit feedback setting, achieving asymptotic and non-asymptotic convergence, respectively. It is worth noting that Cui et al. (2022) proposed the first algorithm that has sample complexity independent of the number of actions.

---

*Equal contribution.

Offline reinforcement learning has been studied in many real-world applications (Levine et al., 2020). From the theoretical perspective, a line of work provides understanding of offline single-agent decision making, including bandits and Markov Decision Processes (MDPs), where researchers derived favorable sample complexity under the single policy coverage (Rashidinejad et al., 2021; Xie et al., 2021b). However, how to learn in offline multi-agent games with offline data is still far from clear. Recently, the unilateral coverage assumption has been proposed as the minimal assumption for offline zero-sum games and offline general-sum games with corresponding algorithms to learn the NE (Cui & Du, 2022a;b; Zhong et al., 2022). Though their coverage assumption and the algorithms apply to the most general class of normal-form games, when specialized to congestion games, the sample complexity will scale with the number of actions, which can be exponentially large. Since congestion games admit specific structures, one may hope to find specialized data coverage assumptions that permit sample-efficient offline learning.

In different applications, the types of feedback, i.e., the revealed reward information, can be different in the offline dataset. For instance, the dataset may include the reward of each facility, the reward of each player, or the total reward of the game. With decreasing information contained in the dataset, different coverage assumptions and algorithms are necessary. In addition, the main challenge in solving congestion games lies in the curse of an exponentially large action set, as the number of actions can be exponential in the number of facilities. In this work, we aim to answer the following question:

|  | One-Unit Deviation | Weak Covariance Domination | Strong Covariance Domination |
|---|---|---|---|
| Facility-Level | ✔ | ✔ | ✔ |
| Agent-Level | ✘ | ✔ | ✔ |
| Game-Level | ✘ | ✘ | ✔ |

Table 1: A summary of how data coverage assumptions affect offline learnability. In particular, ✔ represents under this pair of feedback type and assumption, an NE can be learned with a sufficient amount of data; on the other hand, ✘ represents there exists some instances in which a NE cannot be learned no matter how much data is collected.

*When can we find approximate NE in offline congestion games with different types of feedback, without suffering from the curse of large action set?*

We provide an answer that reveals striking differences between different types of feedback.

## 1.1 MAIN CONTRIBUTIONS

We provide both positive and negative results for each type of feedback. See Table 1 for a summary.

**1. Three types of feedback and corresponding data coverage assumptions**. We consider three types of feedback: facility-level feedback, agent-level feedback and game-level feedback to model different real-world applications and what dataset coverage assumptions permit finding an approximate NE. In offline general-sum games, Cui & Du (2022b) proposes the unilateral coverage assumption. Although their result can be applied to offline congestion games with agent-level feedback, their unilateral coverage coefficient is at least as large as the number of actions and thus has an exponential dependence on the number of facilities. Therefore, for each type of feedback, we propose a corresponding data coverage assumption to escape the curse of the large action set. Specifically:

• **Facility-Level Feedback**: For facility-level feedback, the reward incurred in each facility is provided in the offline dataset. This type of feedback has the strongest signal. We propose the One-Unit Deviation coverage assumption (cf. Assumption 2) for this feedback.

• **Agent-Level Feedback**: For agent-level feedback, only the sum of the facilities' rewards for each agent is observed. This type of feedback has weaker signals than the facility-level feedback does, and therefore we require a stronger data coverage assumption (cf. Assumption 4).

• **Game-Level Feedback**: For the game-level feedback, only the sum of the agent rewards is obtained. This type of feedback has the weakest signals, and we require the strongest data coverage assumption (Assumption 5).

Notably, for the latter two types of feedback, we leverage the connections between congestion games and linear bandits.

**2. Sample complexity analyses for different types of feedback**. We adopt the surrogate minimization idea in Cui & Du (2022b) and show a unified algorithm (cf. Algorithm 1) with carefully designed bonus terms tailored to different types of feedback can efficiently find an approximate NE, therefore showing our proposed data coverage assumptions are *sufficient*. For each type of feedback, we give a polynomial upper bound under its corresponding dataset coverage assumption.

**3. Separations between different types of feedback.** To rigorously quantify the signal strengths in the three types of feedback, we provide concrete hard instances. Specifically, we show there exists a problem instance that satisfies Assumption 2, but with only agent-level feedback, we *provably cannot* find an approximate NE, yielding a separation between the facility-level feedback and the agent-level feedback. Furthermore, we also show there exists a problem instance that satisfies Assumption 4, but with only game-level feedback, we *provably cannot* find an approximate NE, yielding a separation between the agent-level feedback and game-level feedback.

In addition, we also provide several concrete scenarios to exemplify and motivate the aforementioned three types of feedback, which can be found in Appendix A.

## 1.2 RELATED WORK

**Potential Games and Congestion Games.** Potential games are a special class of normal-form games with a potential function to quantify the changes in the payoff of each player and deterministic NE is proven to exist (Monderer & Shapley, 1996). Asymptotic convergence to the NE can be achieved by classic game theory dynamics such as best response dynamic (Durand, 2018; Swenson et al., 2018), replicator dynamic (Sandholm et al., 2008; Panageas & Piliouras, 2016) and no-regret dynamic (Heliou et al., 2017). Recently, Cen et al. (2021) proved that natural policy gradient has a convergence rate independent of the number of actions in entropy regularized potential games. Anagnostides et al. (2022) provided the non-asymptotic convergence rate for mirror descent and $O(1)$ individual regret for optimistic mirror descent.

Congestion games are proposed in the seminal work (Rosenthal, 1973) and the equivalence with potential games is proven in (Monderer & Shapley, 1996). Note that congestion games can have exponentially large action sets, so efficient algorithms for potential games are not necessarily efficient for congestion games. Non-atomic congestion games consider separable players, which enjoy a convex potential function if the cost function is non-decreasing (Roughgarden & Tardos, 2004). For atomic congestion games, the potential function is usually non-convex, making the problem more difficult. (Kleinberg et al., 2009; Krichene et al., 2014) show that no-regret algorithms asymptotically converge to NE with full information feedback and (Krichene et al., 2015) provide averaged iterate convergence for bandit feedback. (Chen & Lu, 2015; 2016) provide non-asymptotic convergence by assuming the atomic congestion game is close to a non-atomic one, and thus approximately enjoys the convex potential. Recently, Cui et al. (2022) first proposed an upper-confidence-bound-type algorithm and a Frank-Wolfe-type algorithm that has a convergence rate without dependence on the number of actions for semi-bandit feedback and bandit feedback settings, respectively. To the best of our knowledge, all of these works either consider the full information setting or the online feedback setting instead of the offline setting in this paper.

**Offline Bandits and Reinforcement Learning.** For related works in empirical offline reinforcement learning, interested readers can refer to (Levine et al., 2020). From the theoretical perspective, researchers have been putting efforts into understanding what dataset coverage assumptions allow for learning the optimal policy. The most basic assumption is the uniform coverage, i.e., every state-action pair is covered by the dataset (Szepesvári & Munos, 2005). Provably efficient algorithms have been proposed for both single-agent and multi-agent reinforcement learning (Yin et al., 2020; 2021; Ren et al., 2021; Sidford et al., 2020; Cui & Yang, 2021; Zhang et al., 2020; Subramanian et al., 2021). In single-agent bandits and reinforcement learning, with the help of pessimism, only single policy coverage is required, i.e., the dataset only needs to cover the optimal policy (Jin et al., 2021; Rashidinejad et al., 2021; Xie et al., 2021b;a). For offline multi-agent Markov games, Cui & Du (2022a) first show that it is impossible to learn with the single policy coverage and they identify the unilateral coverage assumption as the minimal coverage assumption while Zhong et al. (2022); Xiong et al. (2022) provide similar results with linear function approximation. Recently, Yan et al. (2022); Cui & Du (2022b) give minimax sample complexity for offline zero-sum Markov games.

In addition, Cui & Du (2022b) proposes an algorithm for offline multi-player general-sum Markov games that do not suffer from the curse of multiagents.

## 2 PRELIMINARY

### 2.1 CONGESTION GAME

**General-Sum Matrix Game.** A general-sum matrix game is defined by a tuple $\mathcal{G} = (\{\mathcal{A}_i\}_{i=1}^m, R)$, where $m$ is the number of players, $\mathcal{A}_i$ is the action space for player $i$ and $R(\cdot|\boldsymbol{a})$ is a distribution over $[0, r_{\max}]^m$ with mean $\boldsymbol{r}(\boldsymbol{a})$. When playing the game, all players simultaneously select actions, constituting joint action $\boldsymbol{a}$ and the reward is sampled as $\boldsymbol{r} \sim R(\cdot|\boldsymbol{a})$, where player $i$ gets reward $r_i$.

Let $\mathcal{A} = \prod_{i=1}^m \mathcal{A}_i$. A joint policy is a distribution $\pi \in \Delta(\mathcal{A})$ while a product policy is $\pi = \bigotimes_{i=1}^m \pi_i$ with $\pi_i \in \Delta(\mathcal{A}_i)$, where $\Delta(\mathcal{X})$ denotes the probability simplex over $\mathcal{X}$. If the players follow a policy $\pi$, their actions are sampled from the distribution $\boldsymbol{a} \sim \pi$. The expected return of player $i$ under some policy $\pi$ is defined as value $V_i^\pi = \mathbb{E}_{\boldsymbol{a} \sim \pi}[r_i(\boldsymbol{a})]$.

Let $\pi_{-i}$ be the joint policy of all the players except for player $i$. The best response of the player $i$ to policy $\pi_{-i}$ is defined as $\pi_i^{\dagger, \pi_{-i}} = \arg\max_{\mu \in \Delta(\mathcal{A}_i)} V_i^{\mu, \pi_{-i}}$. Here $\mu$ is the policy for player $i$ and $(\mu, \pi_{-i})$ constitutes a joint policy for all players. We can always set the best response to be a pure strategy since the value function is linear in $\mu$. We also denote $V_i^{\dagger, \pi_{-i}} := V_i^{\pi_i^{\dagger, \pi_{-i}}, \pi_{-i}}$ as the best response value. To evaluate a policy $\pi$, we use the performance gap defined as

$$\text{Gap}(\pi) = \max_{i \in [m]} \left[ V_i^{\dagger, \pi_{-i}} - V_i^\pi \right].$$

A product policy $\pi$ is an $\varepsilon$-approximate NE if $\text{Gap}(\pi) \leq \varepsilon$. A product policy $\pi$ is an NE if $\text{Gap}(\pi) = 0$. Note that it is possible to have multiple Nash equilibria in one game.

**Congestion Game.** A congestion game is a general-sum matrix game with special structures. In particular, there is a facility set $\mathcal{F}$ such that $a \subseteq \mathcal{F}$ for all $a \in \mathcal{A}_i$, meaning that the size of $\mathcal{A}_i$ can at most be $2^F$, where $F = |\mathcal{F}|$. A set of facility reward distributions $\{R^f(\cdot|n) | n \in \mathbb{N}\}_{f \in \mathcal{F}}$ is associated with each facility $f$. Let the number of players choosing facility $f$ in the action be $n^f(\boldsymbol{a}) = \sum_{i=1}^m \mathbb{1}\{f \in a_i\}$, where $\boldsymbol{a}$ is the joint action. A facility with specific number of players selecting it is said to be a configuration on $f$. Two joint actions where the same number of players selecting $f$ are said to have the same configuration on $f$. The reward associated with facility $f$ is sampled by $r^f \sim R^f(\cdot|n^f(\boldsymbol{a}))$ and the total reward of player $i$ is $r_i = \sum_{f \in a_i} r^f$. With slight abuse of notation, let $r^f(n)$ be the mean reward that facility $f$ generates when there are $n$ players choosing it. We further assume the support of $R^f(\cdot|n)$ is [-1,1] for all $n \in [m]$. It is well known that every congestion game has pure strategy NE.

The information we get from the game each episode is $(\boldsymbol{a}^k, \boldsymbol{r}^k)$, where $\boldsymbol{a}^k$ is the joint action and $\boldsymbol{r}^k$ contains the reward signal. In this paper, we will consider three types of reward feedback in congestion games, which essentially make $\boldsymbol{r}^k$ different in each data point $(\boldsymbol{a}^k, \boldsymbol{r}^k)$.

• **Facility-Level Feedback (Semi-Bandit Feedback):** In each data point $(\boldsymbol{a}^k, \boldsymbol{r}^k)$, $\boldsymbol{r}^k$ contains reward received from each facility $f \in \bigcup_{i=1}^m a_i^k$, meaning that $\boldsymbol{r}^k = \{r^{f,k}\}_{f \in \bigcup_{i=1}^m a_i^k}$.

• **Agent-Level Feedback (Bandit Feedback):** In each data point $(\boldsymbol{a}^k, \boldsymbol{r}^k)$, $\boldsymbol{r}^k$ contains reward received by each player, meaning that $\boldsymbol{r}^k = \{r_i^k\}_{i=1}^m$, where $r_i^k = \sum_{f \in a_i^k} r^{f,k}$.

• **Game-Level Feedback:** In each data point $(\boldsymbol{a}^k, \boldsymbol{r}^k)$, $\boldsymbol{r}^k$ contains only the total reward received by all players, meaning that $\boldsymbol{r}^k = \sum_{i=1}^m r_i^k$, which becomes a scalar. This type of feedback is the minimal information we can get and has not been discussed in previous literature.

### 2.2 OFFLINE MATRIX GAME

**Offline Matrix Game.** In the offline setting, the algorithm only has access to an offline dataset $\mathcal{D} = \{(\boldsymbol{a}^k, \boldsymbol{r}^k)\}_{k=1}^n$ collected by some exploration policy $\rho$ in advance.

A joint action $\boldsymbol{a}$ is said to be covered if $\rho(\boldsymbol{a}) > 0$. Cui & Du (2022a) has proved that the following assumption is a minimal dataset coverage assumption to learn an NE in a general-sum matrix game. The assumption requires the dataset to cover all unilaterally deviated actions from one NE.

**Assumption 1.** *There exists an NE $\pi^*$ such that for any player $i$ and policy $\pi_i \in \Delta(\mathcal{A})$, $\boldsymbol{a}$ is covered by $\rho$ as long as $(\pi_i, \pi^*_{-i})(\boldsymbol{a}) > 0$.*

Cui & Du (2022b) provides a sample complexity result for matrix games with dependence on $C(\pi^*)$, where $C(\pi)$ quantifies how well $\pi$ is unilaterally covered by the dataset. The definition is as follows.

**Definition 1.** *For strategy $\pi$ and $\rho$ satisfying Assumption 1, the unilateral coefficient is defined as*

$$C(\pi) = \max_{i, \pi', \rho(\boldsymbol{a}) > 0} \frac{(\pi'_i, \pi_{-i})(\boldsymbol{a})}{\rho(\boldsymbol{a})}. \tag{1}$$

**Surrogate Minimization.** Cui & Du (2022b) proposed an algorithm called Strategy-wise Bonus + Surrogate Minimization (SBSM) to achieve efficient learning under Assumption 1. SBSM motivates a general algorithm framework for learning congestion games in different settings. First we design $\widehat{r}_i(\boldsymbol{a})$ which estimates the reward player $i$ gets when the joint action is $\boldsymbol{a}$. Offline bandit (reinforcement learning) algorithm usually leverages the confidence bound (bonus) to create a pessimistic estimate of the reward estimator, inducing conservatism in the output policy and achieving a sample-efficient algorithm. Here we formally define bonus as follows.

**Definition 2.** *For any reward estimator $\widehat{r}_i : \mathcal{A} \to \mathbb{R}$ that estimates reward with expectation $r_i : \mathcal{A} \to \mathbb{R}$, $b_i : \mathcal{A} \to \mathbb{R}$ is called the bonus term for $\widehat{r}$ if for all $i \in [m], \boldsymbol{a} \in \mathcal{A}$, with probability at least $1 - \delta$, it holds that*

$$|r_i(\boldsymbol{a}) - \widehat{r}_i(\boldsymbol{a})| \le b_i(\boldsymbol{a}). \tag{2}$$

The formulae for $\widehat{r}_i$ and $b$ vary according to the type of feedback as discussed in later sections. The optimistic and pessimistic values for policy $\pi$ and player $i$ are defined as

$$\overline{V}_i^\pi = \mathbb{E}_{\boldsymbol{a} \sim \pi} [\widehat{r}_i(\boldsymbol{a}) + b_i(\boldsymbol{a})], \quad \underline{V}_i^\pi = \mathbb{E}_{\boldsymbol{a} \sim \pi} [\widehat{r}_i(\boldsymbol{a}) - b_i(\boldsymbol{a})]. \tag{3}$$

Finally, the algorithm minimizes $\max_{i \in [m]} \left[ \overline{V}_i^{\dagger, \pi_{-i}} - \underline{V}_i^\pi \right]$ over the policy $\pi$, which serves as a surrogate of the performance gap (see Lemma 1). We summarize it in Algorithm 1. Note that we only take the surrogate gap from SBSM but not the strategy-wise bonus, which is a deliberately designed bonus term depending on the policy. Instead, we design specialized bonus terms by exploiting the unique structure of the congestion game, which will be discussed in detail in later sections.

---

**Algorithm 1** Surrogate Minimization for Congestion Games

---

**Require:** Offline dataset $\mathcal{D}$
 1: Compute $\widehat{r}(\boldsymbol{a}), b(\boldsymbol{a})$ for all $\boldsymbol{a} \in \mathcal{A}$ according to the dataset $\mathcal{D}$.
 2: Compute the optimistic value $\overline{V}_i^\pi$ and pessimistic value $\underline{V}_i^\pi$ for all policy $\pi$ and player $i$ by (3).
 3: Compute $\overline{V}_i^{\dagger, \pi_{-i}} = \max_{\pi'_i \in \Delta(\mathcal{A}_i)} \overline{V}_i^{\pi'_i, \pi_{-i}}$.
 4: **return** $\arg \min_\pi \max_{i \in [m]} \left[ \overline{V}_i^{\dagger, \pi_{-i}} - \underline{V}_i^\pi \right]$.

---

The sample complexity of this algorithm is guaranteed by the following theorem.

**Theorem 1.** *Let $\Pi$ be the set of all deterministic policies and $b$ is a bonus term for $\widehat{r}$. With probability $1 - \delta$, it holds that*

$$Gap(\pi^{output}) \le 2 \max_{i \in [m]} \left[ \max_{\pi' \in \Pi} \mathbb{E}_{\boldsymbol{a} \sim (\pi'_i, \pi^*_{-i})} b_i(\boldsymbol{a}) + \mathbb{E}_{\boldsymbol{a} \sim \pi^*} b_i(\boldsymbol{a}) \right].$$

*where $\pi^{output}$ is the output of Algorithm 1.*

Here, the expectation of bonus term over some policy reflects the degree of uncertainty of the reward under that policy. Inside the operation $\min_{\pi \in \Pi}[\cdot]$, the first term is for unilaterally deviated policy from $\pi$ that maximizes this uncertainty and the second term is the uncertainty for $\pi$. The full proof is deferred to Appendix B. This theorem tells us that if we want to bound the performance gap, we need to precisely estimate rewards induced by unilaterally deviated actions from the NE, which caters to Assumption 1.

## 3 Offline Congestion Game with Facility-level Feedback

Recall that for Definition 1. If $\pi$ is deterministic, the minimum value of $C(\pi)$ is achieved when $\rho$ uniformly covers all actions achievable by unilaterally deviating from $\pi$ (see Proposition 1). Since having $\pi'_i$ deviating from $\pi$ can at least cover $A_i$ actions, the smallest value of $C(\pi)$ scales with $\max_{i \in [m]} A_i$, which is reasonable for general-sum matrix game. However, this is not acceptable for congestion games since the size of action space can be exponential ($A_i \leq 2^F$). As a result, covering all possible unilateral deviations becomes inappropriate.

Compared to general-sum games, congestion games with facility-level feedback inform us not only the total reward but also the individual rewards from all chosen facilities. This allows us to estimate the reward distribution from each facility separately. Instead of covering all unilaterally deviating actions $\boldsymbol{a}$, we only need to make sure for any such action $\boldsymbol{a}$ and any facility $f \in \mathcal{F}$, we cover some actions that share the same configuration with $\boldsymbol{a}$ on $f$. This motivates the dataset coverage assumption on facilities rather than actions. In particular, we quantify the facility coverage condition and present the new assumption as follows.

**Definition 3.** *For strategy $\pi$, facility $f$ and integer $n$, the facility cumulative density is defined as*

$$d_f^\pi(n) = \sum_{\boldsymbol{a}:n^f(\boldsymbol{a})=n} \pi(\boldsymbol{a}).$$

Figure 1: Illustration of Assumption 2. There are five facilities and five players with full action space. The facility configuration in $\pi^*$ is marked in red. The transparent boxes cover the facility configuration required in the assumption.

*Furthermore, a facility $f$ is said to be covered by $\rho$ at $n$ if $d_f^\rho(n) > 0$.*

**Assumption 2** (One-Unit Deviation). *There exists an NE $\pi^*$ such that for any player $i$, facility $f$ and integer $n$, if there exists a policy $\pi_i \in \Delta(\mathcal{A})$ such that $d_f^{\pi_i, \pi^*_{-i}}(n) > 0$, we have $d_f^\rho(n) > 0$.*

In plain text, this assumption requires us to cover all possible facility configurations induced by unilaterally deviated actions. As mentioned in Section 2.1, we can always choose $\pi^*$ to be deterministic. In the view of each facility, the unilateral deviation is either a player who did not select it now selects it or a player who selected it now does not select it. Thus for each $f \in \mathcal{F}$, it is sufficient to cover configurations where the number of players selecting $f$ differs from that number of NE by 1. This is why we call it the one-unit deviation assumption. The facility coverage condition is visualized in Figure 1. Meanwhile, Definition 1 is adapted to this assumption as follows.

**Assumption 3.** *There exists a constant $C_{\text{facility}} > 0$ and an NE $\pi^*$ such that $C_{\text{facility}} \geq \max_{i,\pi,f} \frac{d_f^{\pi_i, \pi^*_{-i}}(n)}{d_f^\rho(n)}$, where we use the convention that $\frac{0}{0} = 0$.*

The sample complexity bound depends on $C_{\text{facility}}$ (see Theorem 3). The minimum value of $C_{\text{facility}}$ is at most 3, which is acceptable (see Proposition 2). Furthermore, we show that no assumption weaker than the one-unit deviation allows NE learning, as stated in the following theorem.

**Theorem 2.** *Define a class $\mathcal{X}$ of congestion game $M$ and exploration strategy $\rho$ that consists of all $M$ and $\rho$ pairs that Assumption 2 is satisfied except for at most one configuration for one facility. For any algorithm **ALG** there exists $(M, \rho) \in \mathcal{X}$ such that the output of **ALG** is at most a $1/2$-NE strategy no matter how much data is collected.*

*Proof Sketch.* Consider congestion games with a single facility $f$ and five players. The action space for each player is $\{\varnothing, \{f\}\}$. We construct the following two congestion games with deterministic rewards. The facility coverage condition of NEs are marked using bold symbols in the table. The exploration policy is set to be

$$\rho(\boldsymbol{a}) = \begin{cases} 1/20 & \text{one, three or four players select } f, \\ 0 & \text{otherwise.} \end{cases}$$

| | | | | | |
|---|---|---|---|---|---|
| Congestion Game 1: | $\boldsymbol{R^f(1)=1}$ | $R^f(2)=-1$ | $R^f(3)=1$ | $R^f(4)=1$ | $\boldsymbol{R^f(5)=1}$ |
| Congestion Game 2: | $R^f(1)=1$ | $R^f(2)=1$ | $R^f(3)=1$ | $\boldsymbol{R^f(4)=1}$ | $R^f(5)=-1$ |

These two games with $\rho$ are not distinguishable for **ALG**. Full proof is deferred to Appendix B. $\square$

In the facility-level feedback setting, the bonus term is similar to that from Cui et al. (2022). First, we count the number of tuples in dataset $\mathcal{D}$ with $n$ players choosing facility $f$ as $N^f(n) = \sum_{\boldsymbol{a}^k \in \mathcal{D}} \mathbb{1}\left\{n^f\left(\boldsymbol{a}^k\right) = n\right\}$. Then, we define the estimated reward function and bonus term as

$$\widehat{r}_i(\boldsymbol{a}) = \sum_{f \in a_i} \frac{\sum_{(\boldsymbol{a}^k, \boldsymbol{r}^k) \in \mathcal{D}} r^{f,k} \mathbb{1}\left\{n^f\left(\boldsymbol{a}^k\right) = n^f(\boldsymbol{a})\right\}}{N^f\left(n^f(\boldsymbol{a})\right) \vee 1}, \quad b_i(\boldsymbol{a}) = \sum_{f \in a_i} \sqrt{\frac{\iota}{N^f\left(n^f(\boldsymbol{a})\right) \vee 1}}.$$

Here $\iota = 2\log(4(m+1)F/\delta)$. The contribution for each term in $b_i$ mimics the bonus terms from the well-known UCB algorithm and the sample complexity bound is provided by the following theorem.

**Theorem 3.** *With probability $1-\delta$, if Assumption 2 is satisfied, it holds that*

$$Gap(\pi^{output}) \leq 8\sqrt{m+1}C_{facility}\iota F/\sqrt{n}.$$

The proof of this theorem involves bounding the expectation of $b$ by exploiting the special structure of congestion game. The actions can be classified by the configuration on one facility. This helps bound the expectation over actions, which is essentially the sum over $\mathcal{A}_i$ actions, by the number of players. Detailed proof is deferred to Section B in the appendix.

## 4 OFFLINE CONGESTION GAME WITH AGENT-LEVEL FEEDBACK

### 4.1 IMPOSSIBILITY RESULT

In the agent-level feedback setting, we no longer have access to rewards provided by individual facilities, so estimating them separately is no longer feasible. From limited actions covered in the dataset, we may not be able to precisely estimate rewards for all unilaterally deviated actions, and thus unable to learn an approximate NE. This observation is formalized in the following theorem.

**Theorem 4.** *Define a class $\mathcal{X}$ of congestion game $M$ and exploration strategy $\rho$ that consists of all $M$ and $\rho$ pairs such that Assumption 2 is satisfied. For agent-level feedback, for any algorithm **ALG** there exists $(M, \rho) \in \mathcal{X}$ such that the output of **ALG** is at most a $1/8$-approximate NE no matter how much data is collected.*

*Proof Sketch.* Consider congestion game with two facilities $f_1, f_2$ and two players. Action space for both players is unlimited. We construct the following two congestion games with deterministic rewards. The facility coverage conditions for these NEs are marked by bold symbols in the tables. The exploration policy $\rho$ is set to be

| | | | | |
|---|---|---|---|---|
| $\boldsymbol{R^{f_1}(2)=1/2}$ | $R^{f_2}(2)=-1$ | | $R^{f_1}(2)=-1/4$ | $R^{f_2}(2)=-1/4$ |
| $R^{f_1}(1)=1$ | $R^{f_2}(1)=1$ | | $\boldsymbol{R^{f_1}(1)=1}$ | $\boldsymbol{R^{f_2}(1)=1}$ |
| Congestion Game 3 | | | Congestion Game 4 | |

$$\rho(a_1, a_2) = \begin{cases} 1/3 & a_1 = a_2 = \{f_1, f_2\}, \\ 1/3 & a_1 = \{f_1\}, a_2 = \{f_2\} \text{ or } a_1 = \{f_2\}, a_2 = \{f_1\} \\ 0 & \text{otherwise.} \end{cases} \quad (4)$$

It can be verified that both $f_1$ and $f_2$ are covered at 1 and 2 as shown in Figure 2. These two games look identical with agent-level feedback. Hence it is impossible for the algorithm to distinguish them. Detailed proof is deferred to Appendix C. $\square$

### 4.2 SOLUTION VIA LINEAR BANDIT

In the agent-level feedback setting, a congestion game can be viewed as $m$ linear bandits. Let $\theta$ be a $d$-dimensional vector where $d = mF$ and $r^f(n) = \theta_{n+mf}$. Let $A_i : \mathcal{A} \to \{0,1\}^d$ and

$$[A_i(\boldsymbol{a})]_j = \mathbb{1}\{j = n + mf, f \in a_i, n = n^f(\boldsymbol{a})\}.$$

Here we assign each facility an index in $0, 1, \cdots, F - 1$. Then the mean reward for player $i$ can be written as $r_i(\boldsymbol{a}) = \langle A_i(\boldsymbol{a}), \theta \rangle$. In the view of bandit problem, $i$ is the index of the bandit and the action taken is $\boldsymbol{a}$, which is identical for all $m$ bandits. $\widehat{r}_i(\boldsymbol{a}) = \langle A_i(\boldsymbol{a}), \widehat{\theta} \rangle$ where $\widehat{\theta}$ can be estimated through ridge regression together with bonus term as follows.

Figure 2: Facility coverage condition for $\rho$. Each pair $(f, n)$ represents the configuration that $n$ players select facility $f$. Each box contains the facility coverage condition for one player. There are two classes of covered actions as described in the formula (4). The color of each box represents the class of actions it belongs to.

$$\widehat{\theta} = V^{-1} \sum_{(\boldsymbol{a}^k, \boldsymbol{r}^k) \in \mathcal{D}} \sum_{i \in [m]} A_i(\boldsymbol{a}^k) r_i^k, \quad V = I + \sum_{(\boldsymbol{a}^k, \boldsymbol{r}^k) \in \mathcal{D}} \sum_{i \in [m]} A_i(\boldsymbol{a}^k) A_i(\boldsymbol{a}^k)^\top. \quad (5)$$

$$b_i(\boldsymbol{a}) = \|A_i(\boldsymbol{a})\|_{V^{-1}} \sqrt{\beta}, \quad \text{where } \sqrt{\beta} = 2\sqrt{d} + \sqrt{d \log\left(1 + \frac{mnF}{d}\right) + \iota}. \quad (6)$$

Jin et al. (2021) studied offline linear Markov Decision Process (MDP) and proposed a sufficient coverage assumption for learning optimal policy. A linear bandit is essentially a linear MDP with only one state and horizon equals to 1. Here we adapt the assumption to bandit setting and generalize it to congestion game in Assumption 4.

**Assumption 4** (Weak Covariance Domination). *There exists a constant $C_{agent} > 0$ and an NE $\pi^*$ such that for all $i \in [m]$ and policy $\pi_i$, it holds that*

$$V \succeq I + nC_{agent}\mathbb{E}_{\boldsymbol{a} \sim (\pi_i, \pi^*_{-i})}\left[A_i(\boldsymbol{a})A_i(\boldsymbol{a})^\top\right]. \quad (7)$$

To see why Assumption 4 implies learnability, notice that the right hand side of (7) is equal to the expectation of the covariance matrix $V$ if the data is collected by running policy $(\pi_j, \pi^*_{-j})$ for $C_{agent}n$ episodes. By using such a matrix, we can estimate the rewards of actions sampled from $(\pi_j, \pi^*_{-j})$ precisely via linear regression. Here, Assumption 4 states that for all unilaterally deviated policy $(\pi_j, \pi^*_{-j})$, we can estimate the rewards it generate at least as well as collecting data from $(\pi_j, \pi^*_{-j})$ for $C_{agent}n$ episodes, which implies that we can learn an approximate NE (see Theorem 1). Under Assumption 4, we can obtain the sample complexity bound as follows.

**Theorem 5.** *If Assumption 4 is satisfied, with probability $1 - \delta$, it holds that*

$$Gap(\pi^{output}) \leq 4\sqrt{\frac{mF\beta}{C_{agent}n}},$$

*where $\sqrt{\beta}$ is defined in (6) and $\pi^{output}$ is the output of Algorithm 1..*

*Remark* 1. As an illustrative example, consider a congestion game and full action space, i.e. $\mathcal{A}_i = 2^{\mathcal{F}}$ for all player $i$ with pure strategy NE. The dataset uniformly covers all actions where only one player deviates and only deviates on one facility. For example, if player 1 chooses $\{f_1, f_2\}$, the dataset should cover player 1 selecting $\{f_1\}, \{f_2\}, \{f_1, f_2, f_3\}, \{f_1, f_2, f_4\}, \cdots$ with other players unchanged. There are $F$ such actions for each player, so the dataset covers $mF$ actions in total. The change in reward when a single player deviates from $\pi^*$ is the sum of change in reward from each deviated facility. With sufficient data, we can precisely estimate the change in reward from each deviated facility and estimate the reward from any unilaterally deviated action afterward. With high probability, $C_{agent}$ for this example is no smaller than $1/2mF^4$ (see Proposition 4 in the appendix). Hence with appropriate dataset coverage, our algorithm can achieve sample-efficient approximate NE learning in agent-level feedback.

## 5 OFFLINE CONGESTION GAME WITH GAME-LEVEL FEEDBACK

With less information revealed in game-level feedback, a stronger assumption is required to learn an approximate NE, which is formally stated in Theorem 6. The proof is similar to that of Theorem 4 and we defer it to the Appendix D.

**Theorem 6.** *Define a class $\mathcal{X}$ of congestion game $M$ and exploration strategy $\rho$ that consists of all $M$ and $\rho$ pairs such that Assumption 4 is satisfied. For game-level feedback, for any algorithm* ***ALG*** *there exists $(M, \rho) \in \mathcal{X}$ such that the output of* ***ALG*** *is at least $1/4$-approximate NE no matter how much data is collected.*

In the game-level feedback setting, a congestion game can be viewed as a linear bandit. Let $A : \mathcal{A} \to \{0, 1\}^d$ and $A(\boldsymbol{a}) = \sum_{i \in [m]} A_i(\boldsymbol{a})$. The game-level reward can be written as $r(\boldsymbol{a}) = \langle A(\boldsymbol{a}), \theta \rangle$. Thus, we can similarly use ridge regression and build bonus terms as follows.

$$\widehat{r}_i(\boldsymbol{a}) = \left\langle A_i(\boldsymbol{a}), \widehat{\theta} \right\rangle, \quad \widehat{\theta} = V^{-1} \sum_{(\boldsymbol{a}^k, r^k) \in \mathcal{D}} A(\boldsymbol{a}^k) r^k, \quad V = I + \sum_{(\boldsymbol{a}^k, r) \in \mathcal{D}} A(\boldsymbol{a}^k) A(\boldsymbol{a}^k)^\top, \quad (8)$$

$$b_i(\boldsymbol{a}) = \max_{i \in [m]} \|A_i(\boldsymbol{a})\|_{V^{-1}} \sqrt{\beta}, \quad \text{where } \sqrt{\beta} = 2\sqrt{d} + \sqrt{d \log (1 + nm) + \iota}. \quad (9)$$

The coverage assumption is adapted from Assumption 4 as follows.

**Assumption 5** (Strong Covariance Domination). *There exists a constant $C_{game} > 0$ and an NE $\pi^*$ such that for all $i \in [m]$ and policy $\pi_i$, it holds that*

$$V \succeq I + nC_{game} \mathbb{E}_{\boldsymbol{a} \sim (\pi_i, \pi^*_{-i})} \left[ A_i(\boldsymbol{a}) A_i(\boldsymbol{a})^\top \right]. \quad (10)$$

Note that although the statement of Assumption 5 is identical to that of Assumption 4, the definition of $V$ has changed, so they are actually different. The interpretation of this assumption is similar to that of Assumption 4. It states that for all unilaterally deviated policy $(\pi_i, \pi^*_{-i})$, we can estimate the reward at least as well as collecting data from $(\pi_i, \pi^*_{-i})$ for $c_{\text{output}} n$ episodes with agent-level feedback. Under this assumption, we get the sample complexity bound as follows.

**Theorem 7.** *If Assumption 5 is satisfied, with probability $1 - \delta$, it holds that*

$$Gap(\pi^{output}) \leq 4 \sqrt{\frac{mF\beta}{C_{game}n}},$$

*where $\beta$ is defined in equation (9) and $\pi^{output}$ is the output of Algorithm 1.*

*Remark* 2. As an illustrative example, consider a congestion game with full action space and pure strategy NE. Let the numbers of players selecting each facility be $(n_1, n_2, \cdots, n_f)$. The dataset uniformly contains the following actions: action where the number of players selecting each facility are $(0, n_2, \cdots, n_f), (n_1 - 1, n_2, \cdots, n_f), (n_1 + 1, n_2, \cdots, n_f)$ and similar actions for other facilities. Besides, we cover an NE action. From this dataset, we can precisely estimate the reward from each single facility with one-unit deviation configuration from NE and hence estimate the reward of unilaterally deviated actions. With high probability, $C_{\text{game}}$ for this example is no smaller than $1/24F^3$ (see Proposition 6 in the appendix). Hence with appropriate dataset coverage, our algorithm can achieve sample-efficient approximate NE learning in game-level feedback.

## 6 CONCLUSION

In this paper, we studied NE learning for congestion games in the offline setting. We analyzed the problem under various types of feedback. Hard instances were constructed to show separations between different types of feedback. For each type of feedback, we identified dataset coverage assumptions to ensure NE learning. With tailored reward estimators and bonus terms, we showed the surrogate minimization algorithm is able to find an approximate NE efficiently.

## ACKNOWLEDGEMENTS

This work was supported in part by NSF TRIPODS II-DMS 2023166, NSF CCF 2007036, NSF IIS 2110170, NSF DMS 2134106, NSF CCF 2212261, NSF IIS 2143493, NSF CCF 2019844.

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

## A  MOTIVATING EXAMPLES

In this section, we provide concrete scenarios for each type of feedback.

*Example* 1 (**Facility-level feedback**). Suppose Google Maps is trying to improve its route assigning algorithm through historical data based on certain regions. Then, each edge (road) on the traffic graph of this region can be considered as a facility and the action that a user will take is a path that connects a certain origin and destination. In this setting, the cost of each facility is the waiting time on that road, which may increase as the number of users choosing this facility increases. In the historical data, each data point contains the path chosen by each user and his/her waiting time on each road, which is an offline dataset with facility-level feedback.

*Example* 2 (**Agent-level feedback**). Suppose a company is trying to learn a policy to advertise its products from historical data. We can consider a certain set of websites as the facility set, and the products as the players. The action chosen for each product is a subset of websites where the company will place advertisements for that product. The reward for each product is measured by its sales. In the historical data, each data point contains the websites chosen for each product advertisement and the total amount of sales within a certain range of time. This offline dataset inherently has only agent-level feedback since the company cannot measure each website's contribution to sales.

*Example* 3 (**Game-level feedback**). Under the same setting above, suppose now another company (called B) is also trying to learn such a policy but lacks internal historical data. Therefore, B decides to use the data from the company mentioned in the above example (called A). However, since company B does not have internal access to company A's database, the precise sales of each product is not visible to company B. As a result, company B can only record the total amount of sales of all concerning products from company A's public financial reports, making its offline dataset have only the game-level feedback.

## B  OMITTED PROOF IN SECTION 3

**Lemma 1.** *With probability $1 - \delta$, for any policy $\pi$, we have*

$$Gap(\pi) \leq \max_{i \in [m]} \left[ \overline{V}_i^{\dagger, \pi_{-i}} - \underline{V}_i^{\pi} \right].$$

*In addition, we have*

$$Gap(\pi^{output}) \leq \min_{\pi} \max_{i \in [m]} \left[ \overline{V}_i^{\dagger, \pi_{-i}} - \underline{V}_i^{\pi} \right].$$

*Proof.* By (3) and (2), with probability $1 - \delta$

$$\underline{V}_i^{\pi} \leq V_i^{\pi} \leq \overline{V}_i^{\pi}.$$

Hence

$$\mathbf{Gap}(\pi) = \max_{\pi'} \max_{i \in [m]} \left[ V_i^{\pi_i', \pi_{-i}} - V_i^{\pi} \right] \leq \max_{\pi'} \max_{i \in [m]} \left[ \overline{V}_i^{\pi_i', \pi_{-i}} - \underline{V}_i^{\pi} \right].$$

Since both $\overline{V}_i^{\pi_i', \pi_{-i}}$ and $\underline{V}_i^{\pi}$ are linear in each entry of $\pi$, the first maximizer on the RHS must correspond to a deterministic policy. This proves the first statement. The second statement is by the fact that the algorithm minimizes the RHS of the first statement. □

**Theorem 1.** *Let $\Pi$ be the set of all deterministic policies and $b$ is a bonus term for $\widehat{r}$. With probability $1 - \delta$, it holds that*

$$Gap(\pi^{output}) \leq 2 \max_{i \in [m]} \left[ \max_{\pi' \in \Pi} \mathbb{E}_{\boldsymbol{a} \sim (\pi_i', \pi_{-i}^*)} b_i(\boldsymbol{a}) + \mathbb{E}_{\boldsymbol{a} \sim \pi^*} b_i(\boldsymbol{a}) \right].$$

*where $\pi^{output}$ is the output of Algorithm 1.*

*Proof.*

$$V_i^{\pi} - \underline{V}_i^{\pi} = \mathbb{E}_{\boldsymbol{a} \sim \pi}[r_i(\boldsymbol{a}) - \widehat{r}_i(\boldsymbol{a}) + b_i(\boldsymbol{a})] \leq 2\mathbb{E}_{\boldsymbol{a} \sim \pi} b_i(\boldsymbol{a})$$

$$\overline{V}_i^\pi - V_i^\pi = \mathbb{E}_{\boldsymbol{a} \sim \pi}[\widehat{r}_i(\boldsymbol{a}) - r_i(\boldsymbol{a}) + b_i(\boldsymbol{a})] \le 2\mathbb{E}_{\boldsymbol{a} \sim \pi} b_i(\boldsymbol{a}).$$

Let $\tilde{\pi} = \arg\max_{\pi'} \max_{i \in [m]} \left[ \overline{V}_i^{\pi_i', \pi_{-i}} - \underline{V}_i^\pi \right]$. By similar argument in Lemma 1 we know that we can always choose $\tilde{\pi} \in \Pi$.

$$\begin{aligned}
\text{Gap}\left(\pi^{\text{output}}\right) &\le \min_\pi \max_{i \in [m]} \left[ \overline{V}_i^{\dagger, \pi_{-i}} - \underline{V}_i^\pi \right] \\
&= \min_\pi \max_{i \in [m]} \left[ \overline{V}_i^{\tilde{\pi}_i, \pi_{-i}} - \underline{V}_i^\pi \right] \\
&\le \min_\pi \max_{i \in [m]} \left[ V_i^{\tilde{\pi}_i, \pi_{-i}} - V_i^\pi + 2\mathbb{E}_{\boldsymbol{a} \sim (\tilde{\pi}_i, \pi_{-i})} b_i(\boldsymbol{a}) + 2\mathbb{E}_{\boldsymbol{a} \sim \pi} b_i(\boldsymbol{a}) \right] \\
&\le \min_\pi \left\{ \max_{i \in [m]} \left[ V_i^{\tilde{\pi}_i, \pi_{-i}} - V_i^\pi \right] + \max_{i \in [m]} \left[ 2\mathbb{E}_{\boldsymbol{a} \sim (\tilde{\pi}_i, \pi_{-i})} b_i(\boldsymbol{a}) + 2\mathbb{E}_{\boldsymbol{a} \sim \pi} b_i(\boldsymbol{a}) \right] \right\} \\
&= \min_\pi \left\{ \text{Gap}(\pi) + \max_{i \in [m]} \left[ 2\max_{\pi' \in \Pi} \mathbb{E}_{\boldsymbol{a} \sim (\pi_i', \pi_{-i})} b_i(\boldsymbol{a}) + 2\mathbb{E}_{\boldsymbol{a} \sim \pi} b_i(\boldsymbol{a}) \right] \right\} \\
&\le \text{Gap}(\pi^*) + \max_{i \in [m]} \left[ 2\max_{\pi' \in \Pi} \mathbb{E}_{\boldsymbol{a} \sim (\pi_i', \pi_{-i}^*)} b_i(\boldsymbol{a}) + 2\mathbb{E}_{\boldsymbol{a} \sim \pi^*} b_i(\boldsymbol{a}) \right] \\
&= 2\max_{i \in [m]} \left[ \max_{\pi' \in \Pi} \mathbb{E}_{\boldsymbol{a} \sim (\pi_i', \pi_{-i}^*)} b_i(\boldsymbol{a}) + \mathbb{E}_{\boldsymbol{a} \sim \pi^*} b_i(\boldsymbol{a}) \right]
\end{aligned}$$

$\square$

**Lemma 2.** *With probability $1 - \delta$, we have*

$$|r_i(\boldsymbol{a}) - \widehat{r}_i(\boldsymbol{a})| \le b_i(\boldsymbol{a}), \quad \frac{1}{N^f(n)} \le \frac{4H\iota}{n d_f^\rho(n)}$$

*for all $\boldsymbol{a} \in \mathcal{A}, i \in [m], f \in \mathcal{F}, n \in [m]$.*

*Proof.*

$$r_i(\boldsymbol{a}) - \widehat{r}_i(\boldsymbol{a}) = \sum_{f \in a_i} \left[ \widehat{r}^f \left( n^f(\boldsymbol{a}) \right) - r^f \left( n^f(\boldsymbol{a}) \right) \right].$$

By Hoeffding's bound and union bound we have

$$\left| \widehat{r}^f \left( n^f(\boldsymbol{a}) \right) - r^f \left( n^f(\boldsymbol{a}) \right) \right| \le \sqrt{\frac{2}{N^f \left( n^f(\boldsymbol{a}) \right)} \log \frac{4(m+1)F}{\delta}}$$

for all $f \in \mathcal{F}, \boldsymbol{a} \in \mathcal{A}$ with probability $1 - \delta/2$. Combine the above inequalities we get the first statement hold with probability $1 - \delta/2$. By lemma A.1 of Xie et al. (2021b), replacing $p$ by $d_f^\rho \left( n^f(\boldsymbol{a}) \right)$ and the union bound we get

$$\frac{1}{N^f(n)} \le \frac{8 \log(2(m+1)F/\delta)}{n d_f^\rho(n)} \le \frac{4\iota}{n d_f^\rho(n)}$$

for all $f \in \mathcal{G}, a \in \mathcal{A}$ with probability $1 - \delta/2$. Finally, the proof is complete by using the union bound. $\square$

**Theorem 3.** *With probability $1 - \delta$, if Assumption 2 is satisfied, it holds that*

$$\text{Gap}(\pi^{\text{output}}) \le 8\sqrt{m+1} C_{\text{facility}} \iota F / \sqrt{n}.$$

*Proof.* We have

$$\begin{aligned}
&\mathbb{E}_{\boldsymbol{a} \sim (\pi_i', \pi_{-i}^*)} b_i(\boldsymbol{a}) \\
&= \sum_{f \in \mathcal{F}} \mathbb{E}_{\boldsymbol{a} \sim (\pi_i', \pi_{-i}^*)} \sqrt{\frac{\iota}{N^f \left( n^f(\boldsymbol{a}) \right) \vee 1}}
\end{aligned}$$

$$\leq C_{\text{facility}} \sum_{f \in \mathcal{F}} \sum_{n'=0}^{m} d_f^{\rho}(n') \sqrt{\frac{4\iota^2}{nd_f^{\rho}(n')}}$$

$$= 2C_{\text{facility}}\iota \sum_{f \in \mathcal{F}} \sum_{n'=0}^{m} \sqrt{\frac{d_f^{\rho}(n')}{n}}$$

$$\leq 2C_{\text{facility}}\iota \sum_{f \in \mathcal{F}} \sqrt{\frac{m+1}{n} \sum_{n'=0}^{m} d_f^{\rho}(n')}$$

$$\leq 2\sqrt{m+1}C_{\text{facility}}\iota F/\sqrt{n}$$

The first inequality is by Definition 3 and Lemma 2. The second inequality is by the fact that $d_f^{\rho}(\boldsymbol{a}) \leq 1$. Combine this with Theorem 1 and Lemma 2 we get the conclusion. $\square$

**Theorem 2.** *Define a class $\mathcal{X}$ of congestion game $M$ and exploration strategy $\rho$ that consists of all $M$ and $\rho$ pairs that Assumption 2 is satisfied except for at most one configuration for one facility. For any algorithm **ALG** there exists $(M, \rho) \in \mathcal{X}$ such that the output of **ALG** is at most a $1/2$-NE strategy no matter how much data is collected.*

*Proof.* Consider congestion game with a single facility $f$ and five players. The action space for each player is $\{\varnothing, \{f\}\}$. We construct the following two congestion games with deterministic rewards. Since there is only one facility, the reward players receive and whether a joint action is NE only depends on the configuration, i.e. the number of players selecting $f$. Hence in the remaining part of the proof we will use configuration to describe the action. For the first game, there are two NEs, which are "only one player selecting $f$" and "all players selecting $f$". For the second game, the NE is "four players selecting $f$". The exploration policy is set to be

$$\rho(\boldsymbol{a}) = \begin{cases} 1/20 & \text{one, three or four players select } f, \\ 0 & \text{otherwise.} \end{cases}$$

For the first game, we cover the first NE and its unilateral deviation except for two players selecting

$$\boldsymbol{R^f(1)} = \boldsymbol{1} \quad R^f(2) = -1 \quad R^f(3) = 1 \quad R^f(4) = 1 \quad \boldsymbol{R^f(5)} = \boldsymbol{1}$$

Congestion Game 1

$$R^f(1) = 1 \quad R^f(2) = 1 \quad R^f(3) = 1 \quad \boldsymbol{R^f(4)} = \boldsymbol{1} \quad R^f(5) = -1$$

Congestion Game 2

$f$. For the second game, we cover the NE except for five players selecting $f$. Hence both game with $\rho$ are in $\mathcal{X}$ and are not distinguishable for **ALG**. Let the probability of the output policy selecting four players choosing $f$ be $p$. Then it is at least $p$-approximate NE for game 1 and $(1-p)$-approximate NE for game 2. In conclusion, there exists $(M, \rho) \in \mathcal{X}$ such that the output of **ALG** is at most a $1/2$-NE strategy no matter how much data is collected. $\square$

### B.1 PURE NASH EQUILIBRIUM

It is well known that pure Nash equilibrium exists for any congestion game (Rosenthal, 1973). Now we restrict our attention to pure Nash equilibrium and show that we can remove the $\sqrt{m}$ factor in Theorem 3. We use $\Pi^{\text{pure}}$ to denote the set of all pure strategies. We modify Assumption 3 and Algorithm 1 for pure strategies.

**Assumption 6.** *There exists a constant $C_{\text{facility}}^{\text{pure}} > 0$ and a pure NE $\pi^*$ such that*

$$C_{\text{facility}}^{\text{pure}} \geq \max_{i, \pi \in \Pi^{\text{pure}}, f} \frac{d_f^{\pi_i, \pi^*_{-i}}(n)}{d_f^{\rho}(n)},$$

*where we use the convention that $\frac{0}{0} = 0$.*

---

**Algorithm 2** Surrogate Minimization for Congestion Games (Pure Strategy)

**Require:** Offline dataset $\mathcal{D}$
1: Compute $\widehat{r}(\boldsymbol{a}), b(\boldsymbol{a})$ for all $\boldsymbol{a} \in \mathcal{A}$ according to the dataset $\mathcal{D}$.
2: Compute the optimistic value $\overline{V}_i^{\pi}$ and pessimistic value $\underline{V}_i^{\pi}$ for all policy $\pi$ and player $i$ by (3).
3: Compute $\overline{V}_i^{\dagger,\pi_{-i}} = \max_{\pi_i' \in \Delta(\mathcal{A}_i)} \overline{V}_i^{\pi_i',\pi_{-i}}$.
4: **return** $\arg\min_{\pi \in \Pi^{\text{pure}}} \max_{i \in [m]} \left[ \overline{V}_i^{\dagger,\pi_{-i}} - \underline{V}_i^{\pi} \right]$.

---

**Theorem 8.** *With probability $1 - \delta$, if Assumption 6 is satisfied, it holds that*
$$Gap(\pi^{output}) \leq 2\sqrt{3}C_{facility}\iota F/\sqrt{n}.$$

*Proof.* We have
$$\mathbb{E}_{\boldsymbol{a} \sim (\pi_i', \pi_{-i}^*)} b_i(\boldsymbol{a})$$
$$= \sum_{f \in \mathcal{F}} \mathbb{E}_{\boldsymbol{a} \sim (\pi_i', \pi_{-i}^*)} \sqrt{\frac{\iota}{N^f \left( n^f(\boldsymbol{a}) \right) \vee 1}}$$
$$\leq C_{\text{facility}}^{\text{pure}} \sum_{f \in \mathcal{F}} \sum_{n' = n^f(\pi^*) - 1}^{n^f(\pi^*) + 1} d_f^\rho(n') \sqrt{\frac{4\iota^2}{n d_f^\rho(n')}}$$
$$= 2 C_{\text{facility}}^{\text{pure}} \iota \sum_{f \in \mathcal{F}} \sum_{n' = n^f(\pi^*) - 1}^{n^f(\pi^*) + 1} \sqrt{\frac{d_f^\rho(n')}{n}}$$
$$\leq 2 C_{\text{facility}}^{\text{pure}} \iota \sum_{f \in \mathcal{F}} \sqrt{\frac{3}{n} \sum_{n' = n^f(\pi^*) - 1}^{n^f(\pi^*) + 1} d_f^\rho(n')}$$
$$\leq 2\sqrt{3} C_{\text{facility}}^{\text{pure}} \iota F/\sqrt{n}$$

The first inequality is by Definition 3 and Lemma 2. The second inequality is by the definition of $C_{\text{facility}}^{\text{pure}}$. Combine this with Theorem 1 and Lemma 2 we get the conclusion. □

### B.2 OMITTED CALCULATIONS IN SECTION 3

**Proposition 1.** *Suppose $\pi$ is a deterministic strategy. For a fixed domain of $\rho$, the value of $C(\pi)$ is the smallest when $\rho$ is uniform over all actions achievable from unilaterally deviating from $\pi$.*

*Proof.* Assume that $\rho$ covers an action $\boldsymbol{a}$ which is not achievable from unilaterally deviating from $pi$, then we construct a new $\rho'$ where $\rho'(\boldsymbol{a}) = 0$ and the other entries scales up by factor $1/(1 - \rho(\boldsymbol{a}))$. $\rho'$ achieves larger $C(\pi)$ than $\rho$. Hence $\rho$ only cover the actions achievable from unilaterally deviating from $\pi$.

Assume that the distribution is not uniform. Since the best response to a pure strategy can always taken to be a pure strategy, the numerator of 1 can always achieve 1 no matter what $\boldsymbol{a}$ is. Let $\boldsymbol{a}^* = \arg\min_{\boldsymbol{a}} \rho(\boldsymbol{a})$, then there exists $\tilde{\boldsymbol{a}}$ such that $\rho(\tilde{\boldsymbol{a}}) > \rho(\boldsymbol{a}^*)$. Construct $\rho'$ such that $\rho'(\boldsymbol{a}^*) = \rho'(\tilde{\boldsymbol{a}}) = (\rho(\boldsymbol{a}^*) + \rho(\tilde{\boldsymbol{a}}))/2$, then $C(\pi)$ would not increase. By contradiction we get the conclusion. □

**Proposition 2.** *The minimum value of $C_{facility}$ is no larger than $3$.*

*Proof.* Consider the case when $\rho$ is a policy that induces uniform coverage on all facility configurations achievable from $\pi^*$. Since at most three configurations are covered for each facility, the minimum value of $d_f^\rho(n)$ is $1/3$. Thus the minimum value of $\tilde{C}(\pi^*)$ is no larger than $3$ □

## C    OMITTED PROOF IN SECTION 4

**Lemma 3.** *With probability $1 - \delta$ we have*

$$|r_i(\boldsymbol{a}) - \widehat{r}_i(\boldsymbol{a})| \leq b_i(\boldsymbol{a})$$

*for all $i \in [m], \boldsymbol{a} \in \mathcal{A}$.*

*Proof.* As a degenerate version of theorem 20.5 of Lattimore & Szepesvári (2020), we have with probability $1 - \delta$ it holds that

$$\left\| \widehat{\theta} - \theta \right\|_V \leq \|\theta\|_2 + \sqrt{\log \det(V) + 2 \log(1/\delta)}.$$

Hence with probability $1 - \delta$

$$
\begin{aligned}
|r_i(\boldsymbol{a}) - \widehat{r}_i(\boldsymbol{a})| &= \left| \left\langle A_i(\boldsymbol{a}), \widehat{\theta} - \theta \right\rangle \right| \\
&\leq \|A_i(\boldsymbol{a})\|_{V^{-1}} \left\| \widehat{\theta} - \theta \right\|_V \\
&\leq \|A_i(\boldsymbol{a})\|_{V^{-1}} \left( \|\theta\|_2 + \sqrt{\log \det(V) + 2 \log(1/\delta)} \right)
\end{aligned}
$$

for all $i \in [m]$ and $\boldsymbol{a} \in \mathcal{A}$. By Lemma 4 of Cui et al. (2022) we have

$$\det(V) \leq \left( 1 + \frac{mnF}{d} \right)^d$$

since by (5) $\|A_i(\boldsymbol{a})\|_2^2 \leq F$. Besides, $\|\theta\|_2 \leq 2\sqrt{d}$. The proof is complete by combining all these and taking $\max_{i \in [m]}$. $\square$

**Theorem 5.** *If Assumption 4 is satisfied, with probability $1 - \delta$, it holds that*

$$Gap(\pi^{output}) \leq 4\sqrt{\frac{mF\beta}{C_{agent}n}},$$

*where $\sqrt{\beta}$ is defined in (6) and $\pi^{output}$ is the output of Algorithm 1..*

*Proof.* We have for all $i \in [m]$

$$
\begin{aligned}
&\mathbb{E}_{\boldsymbol{a} \sim (\pi_i, \pi_{-i}^*)} b_i(\boldsymbol{a}) \\
=&\mathbb{E}_{\boldsymbol{a} \sim (\pi_i, \pi_{-i}^*)} \sqrt{A_i^\top(\boldsymbol{a}) V^{-1} A_i(\boldsymbol{a})} \\
\leq&\mathbb{E}_{\boldsymbol{a} \sim (\pi_i, \pi_{-i}^*)} \sqrt{A_j^\top(\boldsymbol{a}) \left( I + C_{\text{agent}} n \mathbb{E}_{\boldsymbol{a}' \sim (\pi_i, \pi_{-i}^*)} [A_i(\boldsymbol{a}') A_i(\boldsymbol{a}')^\top] \right)^{-1} A_i(\boldsymbol{a})} \\
=&\mathbb{E}_{\boldsymbol{a} \sim (\pi_i, \pi_{-i}^*)} \sqrt{\text{tr} \left[ \left( I + C_{\text{agent}} n \mathbb{E}_{\boldsymbol{a}' \sim (\pi_i, \pi_{-i}^*)} [A_i(\boldsymbol{a}') A_i(\boldsymbol{a}')^\top] \right)^{-1} A_i(\boldsymbol{a}) A_i^\top(\boldsymbol{a}) \right]} \\
\leq&\mathbb{E}_{\boldsymbol{a} \sim (\pi_i, \pi_{-i}^*)} \sqrt{\text{tr} \left[ \left( I + C_{\text{agent}} n \mathbb{E}_{\boldsymbol{a}' \sim (\pi_i, \pi_{-i}^*)} [A_i(\boldsymbol{a}') A_i(\boldsymbol{a}')^\top] \right)^{-1} A_i(\boldsymbol{a}) A_i^\top(\boldsymbol{a}) \right]} \\
\leq&\sqrt{\text{tr} \left[ \left( I + C_{\text{agent}} n \mathbb{E}_{\boldsymbol{a}' \sim (\pi_i, \pi_{-i}^*)} [A_i(\boldsymbol{a}') A_i(\boldsymbol{a}')^\top] \right)^{-1} \mathbb{E}_{\boldsymbol{a} \sim (\pi_i, \pi_{-i}^*)} A_i(\boldsymbol{a}) A_i^\top(\boldsymbol{a}) \right]} \\
=&\frac{1}{\sqrt{C_{\text{agent}} n}} \sqrt{\text{tr} \left[ I - \left( I + C_{\text{agent}} n \mathbb{E}_{\boldsymbol{a}' \sim (\pi_i, \pi_{-i}^*)} [A_i(\boldsymbol{a}') A_i(\boldsymbol{a}')^\top] \right)^{-1} \right]} \\
\leq&\sqrt{\frac{d}{C_{\text{agent}} n}} = \sqrt{\frac{mF}{C_{\text{agent}} n}}
\end{aligned}
$$

Combine this with Theorem 1 we get the conclusion. $\square$

**Theorem 4.** *Define a class $\mathcal{X}$ of congestion game $M$ and exploration strategy $\rho$ that consists of all $M$ and $\rho$ pairs such that Assumption 2 is satisfied. For agent-level feedback, for any algorithm **ALG** there exists $(M, \rho) \in \mathcal{X}$ such that the output of **ALG** is at most a $1/8$-approximate NE no matter how much data is collected.*

*Proof.* Consider congestion game with two facilities $f_1, f_2$ and two players. Action space for both players are unlimited, i.e. $\mathcal{A}_1 = \mathcal{A}_2 = \{\{f_1\}, \{f_2\}, \{f_1, f_2\}\}$. We construct the following two congestion games with deterministic rewards. The NE for game 3 is $a_1 = \{f_1, f_2\}, a_2 = \{f_1\}$ or $a_2 = \{f_1, f_2\}, a_1 = \{f_1\}$. The NE for game 4 is $a_1 = \{f_1\}, a_2 = \{f_2\}$ or $a_2 = \{f_1\}, a_1 = \{f_2\}$. The facility coverage conditions for these NEs are marked by bold symbols in the tables. The

$$\boldsymbol{R^{f_1}(2) = 1/2} \quad R^{f_2}(2) = -1 \qquad\qquad R^{f_1}(2) = -1/4 \quad R^{f_2}(2) = -1/4$$
$$R^{f_1}(1) = 1 \quad \boldsymbol{R^{f_2}(1) = 1} \qquad\qquad \boldsymbol{R^{f_1}(1) = 1} \qquad \boldsymbol{R^{f_2}(1) = 1}$$

Congestion Game 3         Congestion Game 4

exploration policy $\rho$ is set to be

$$\rho(a_1, a_2) = \begin{cases} 1/3 & a_1 = a_2 = \{f_1, f_2\} \\ 1/3 & a_1 = \{f_1\}, a_2 = \{f_2\} \text{ or } a_1 = \{f_2\}, a_2 = \{f_1\} \\ 0 & \text{otherwise} \end{cases} \tag{11}$$

| $(f_1, 2)$ | $(f_2, 2)$ |
|---|---|
| $(f_1, 1)$ | $(f_2, 1)$ |

Figure 3: Facility coverage condition for $\rho$. Each pair $(f, n)$ represents the configuration that $n$ players select facility $f$. Each box contains the facility coverage condition for one player. There are two classes of covered actions as described in formula (11). The color of each box represents the class of actions it belongs to.

It can be easily verified that both $f_1$ and $f_2$ are covered at 1 and 2. However, all information we may extract from the dataset is $R^{f_1}(1) = 1$, $R^{f_2}(1) = 1$ and $R^{f_1}(2) + R^{f_2}(2) = -1/2$. It is impossible for the algorithm to distinguish these two games. Suppose the output strategy of **ALG** selects action such that two players select $f_1$ with probability $p$. Then $\pi$ is at least a $(1-p)/4$-approximate NE for the first game and at least a $p/4$-approximate NE for the second game. In conclusion, there exists $(M, \rho) \in \mathcal{X}$ such that the output of the algorithm **ALG** is at most a $1/8$-approximate NE strategy no matter how much data is collected. $\qquad\square$

## C.1   OMITTED CALCULATIONS IN SECTION 4

**Lemma 4.** *If $n \geq 8\log((mF+1)/\delta)(mF+1)$, with probability $1 - \delta$, $N(\boldsymbol{a}) \geq \rho(\boldsymbol{a})n/2 = n/2(mF+1)$ for all $\boldsymbol{a}$ with $\rho(\boldsymbol{a}) > 0$.*

*Proof.* $N(\boldsymbol{a})$ follows binomial distribution with parameters $n$ and $\rho(\boldsymbol{a})$. By Chernoff bound, for all $\varepsilon \in \mathbb{R}^+$,

$$\Pr\{N(\boldsymbol{a}) \leq (1-\varepsilon)n\rho(\boldsymbol{a})\} \leq \exp\left(-\frac{\varepsilon^2 n\rho(\boldsymbol{a})}{2}\right)$$

Hence if $\rho \geq -8\log\delta/n$, we have for all $\boldsymbol{a}$ covered in the example, we have

$$\Pr\{N(\boldsymbol{a}) \geq (1-\varepsilon)n\rho(\boldsymbol{a})\} \geq 1 - \exp(-\mu/8) \geq 1 - \delta.$$

By construction $\rho(\boldsymbol{a}) = 1/(mF+1)$ for covered action $\boldsymbol{a}$. By the union bound we get the conclusion. $\qquad\square$

**Proposition 3.** *If $n \geq F$, the maximum value of $C_{agent}$ is no smaller than $1/2F$.*

*Proof.* Let the NE be pure strategy taking joint action $\boldsymbol{a}^*$. For some specific $i$, Let $\mathcal{A}_i^* = \{A_i(a_i, a_{-i}^*) | a_i \in \mathcal{A}_i\}$ and $\mathrm{span}(\mathcal{A}_i^*) = d$, by Kiefer-Wolfowitz theorem there is a joint policy $\hat{\pi}^i$ such that

$$\max_{A \in \mathcal{A}_i^*} A^\top \left[\mathbb{E}_{\boldsymbol{a} \sim \hat{\pi}^i}(A_i(\boldsymbol{a})A_i(\boldsymbol{a}))\right]^{-1} A = d.$$

From now on we scale each entry of $\hat{\pi}^i$ by $1/m$. It is no longer a valid probability measure but the definition of expectation retains. Hence for each $A \in \mathcal{A}_i^*$

$$A^\top \left[ I/m + n\mathbb{E}_{\boldsymbol{a} \sim \hat{\pi}^i} \left( A_i(\boldsymbol{a}) A_i^\top(\boldsymbol{a}) \right) \right]^{-1} A \leq \frac{d}{mn}.$$

By Cauchy inequality for each $A \in \mathcal{A}_i^*$

$$A^\top \left[ I/m + n\mathbb{E}_{\boldsymbol{a} \sim \hat{\pi}^i} \left( A_i(\boldsymbol{a}) A_i^\top(\boldsymbol{a}) \right) \right] A \geq \frac{mn|A|^4}{d}.$$

Let $C = m/d - 1/n$, then for each $A \in \mathcal{A}_i^*$

$$A^\top \left[ I/m + n\mathbb{E}_{\boldsymbol{a} \sim \hat{\pi}^i} \left( A_i(\boldsymbol{a}) A_i(\boldsymbol{a}) \right) \right] A \geq (nC + 1)|A|^4 \geq A^\top (I + nCAA^\top)A.$$

Let $\hat{\pi}$ be the sum of $\hat{\pi}^i$ over all player $i$, then for each $A \in \mathcal{A}_i^*$ and $i$

$$A^\top \left[ I + n\mathbb{E}_{\boldsymbol{a} \sim \hat{\pi}^i} \left( A_i(\boldsymbol{a}) A_i(\boldsymbol{a}) \right) \right] A \geq A^\top (I + nCAA^\top)A.$$

Hence for each $A \in \mathcal{A}_i^*$ and $i$

$$I + n\mathbb{E}_{\boldsymbol{a} \sim \hat{\pi}^i} \left( A_i(\boldsymbol{a}) A_i(\boldsymbol{a}) \right) \succeq I + nCAA^\top.$$

Furthermore, for each $A \in \mathcal{A}_i^*$ and $i$

$$I + n\mathbb{E}_{\boldsymbol{a} \sim \hat{\pi}^i} \left[ A_i(\boldsymbol{a}) A_i(\boldsymbol{a}) \right] \succeq I + nC\mathbb{E}_{\boldsymbol{a} \sim \left( \pi_i, \pi_{-i}^* \right)} \left[ A_i(\boldsymbol{a}) A_i(\boldsymbol{a})^\top \right].$$

Now we are ready to lower bound $C_{\text{agent}}$. By $d \leq mF$

$$C_{\text{agent}} \geq C \geq \frac{m}{2d} \geq \frac{1}{2F}.$$

$\square$

**Proposition 4.** *If $n \geq 8 \log((mF+1)/\delta)(mF+1)$ and $C_{agent} = 1/2mF^4$, then with probability $1 - \delta$, Assumption 4 holds for the example described in Remark 1.*

*Proof.* It suffices to show that inequality 7 holds for all pure strategy $\pi$ and $i$ because the right hand side is linear in any entry of $\pi_i$. From now on, let us focus on some specific $i$ and pure strategy $(\pi_i, \pi_{-i}^*)$ choosing $\tilde{\boldsymbol{a}}$ deterministically.

Without loss of generality, suppose among all elements in $\tilde{\boldsymbol{a}}$, facility that deviates from NE are $f_1, f_2, \cdots, f_s$. For convenience, let $A_0 = A_i(\boldsymbol{a}^*)$ where $\boldsymbol{a}^*$ is the NE. For $f_j$, to estimate its contribution to the reward change, we need an action besides $\boldsymbol{a}^*$, which we denote as $\boldsymbol{a}^{f_j}$. That is, $\boldsymbol{a}^{f_j}$ deviates from $\boldsymbol{a}^*$ only on $f_j$ and let $A_j = A_i(\boldsymbol{a}^{f_j})$. Without loss of generality, suppose the contribution corresponds to $\langle A_j - A_0, \theta \rangle$. Then we can write

$$A_i(\tilde{\boldsymbol{a}}) = A_0 + \sum_{j \in [s]} (A_j - A_0) = \sum_{j \in [s]} A_j + (1 - s)A_0$$

By Lemma 4, it suffices to show

$$I + \frac{n}{2(mF+1)} \sum_{j \in [s]} A_j A_j^\top + \frac{n}{2(mF+1)} A_0 A_0^\top \succeq I + C_{\text{agent}} n A_i(\tilde{\boldsymbol{a}}) A_i(\tilde{\boldsymbol{a}})^\top.$$

In other words, for any $x \in \mathbb{R}^{mF}$, we have

$$\frac{n}{2(mF+1)} \sum_{j \in [s]} x^\top A_j A_j^\top x + \frac{n}{2(mF+1)} x^\top A_0 A_0^\top x \geq C_{\text{agent}} n x^\top A_i(\tilde{\boldsymbol{a}}) A_i(\tilde{\boldsymbol{a}})^\top x.$$

For convenience, let $x_i = x^\top A_i$, this inequality can be rewritten as

$$\frac{n}{2(mF+1)} \sum_{j \in [s]} x_j^2 + \frac{n}{2(mF+1)} x_0^2 \geq C_{\text{agent}} n \left[ \sum_{j \in [s]} x_j + (1 - s)x_0 \right]^2.$$

By Jensen's inequality it suffices to show

$$\frac{n}{2(mF+1)} \sum_{j \in [s]} x_j^2 + \frac{n}{2(mF+1)} x_0^2 \geq C_{\text{agent}} n(s+1) \left[ \sum_{j \in [s]} x_j^2 + (1 - s)^2 x_0^2 \right].$$

Hence it suffices to show

$$\frac{n}{2(mF+1)} \geq C_{\text{agent}} n(F+1)(F-1)^2$$

$\square$

# D    OMITTED PROOF IN SECTION 5

**Lemma 5.** *With probability $1 - \delta$ we have*

$$|r_i(\boldsymbol{a}) - \widehat{r}_i(\boldsymbol{a})| \leq b_i(\boldsymbol{a})$$

*for all $i \in [m], \boldsymbol{a} \in \mathcal{A}$.*

*Proof.* Similar to Lemma 3, we have

$$|r_i(\boldsymbol{a}) - \widehat{r}_i(\boldsymbol{a})| \leq \|A_i(\boldsymbol{a})\|_{V^{-1}} \left( \|\theta\|_2 + \sqrt{\log\det(V) + 2\log(1/\delta)} \right).$$

The bound of $\det(V)$ is now as follows. The basic idea is the same as that of lemma 4 by Cui et al. (2022)

$$\det(V) \leq \left( \frac{\operatorname{tr}(V)}{d} \right)^d = \left( \frac{\operatorname{tr}(I) + \sum_{(\boldsymbol{a}^k, r) \in \mathcal{D}} \|A(\boldsymbol{a}^k)\|_2^2}{d} \right)^d \leq \left( \frac{d + nm^2 F}{d} \right)^d = (1 + nm)^{mF}$$

Besides, $\|\theta\|_2 \leq 2\sqrt{d}$. Combine all these we get the conclusion. $\qquad\square$

**Theorem 7.** *If Assumption 5 is satisfied, with probability $1 - \delta$, it holds that*

$$Gap(\pi^{output}) \leq 4\sqrt{\frac{mF\beta}{C_{game}n}},$$

*where $\beta$ is defined in equation (9) and $\pi^{output}$ is the output of Algorithm 1.*

*Proof.* The proof is identical to that of Theorem 5 except that $C_{game}$ is used instead of $C_{game}$. $\qquad\square$

**Theorem 6.** *Define a class $\mathcal{X}$ of congestion game $M$ and exploration strategy $\rho$ that consists of all $M$ and $\rho$ pairs such that Assumption 4 is satisfied. For game-level feedback, for any algorithm **ALG** there exists $(M, \rho) \in \mathcal{X}$ such that the output of **ALG** is at least $1/4$-approximate NE no matter how much data is collected.*

*Proof.* Similar to the proof of Theorem 4, consider a congestion game with two facilities $f_1, f_2$ and two players. Action space for both players are unlimited, i.e. $\mathcal{A}_1 = \mathcal{A}_2 = \{\{f_1\}, \{f_2\}, \{f_1, f_2\}\}$. We construct the following two congestion games with deterministic rewards. The NE for game 5 is $a_1 = \{f_1\}, a_2 = \{f_1\}$. The NE for game 6 is $a_1 = \{f_1\}, a_2 = \{f_2\}$ or $a_2 = \{f_1\}, a_1 = \{f_2\}$. The exploration policy is set to be

$$
\begin{array}{llll}
\boldsymbol{R^{f_1}(2) = 1/2} & R^{f_2}(2) = -1 & \qquad & R^{f_1}(2) = -1/2 & R^{f_2}(2) = -1 \\
R^{f_1}(1) = 3 & R^{f_2}(1) = -1 & \qquad & \boldsymbol{R^{f_1}(1) = 1} & \boldsymbol{R^{f_2}(1) = 1}
\end{array}
$$

Congestion Game 5       Congestion Game 6

$$\rho(a_1, a_2) = \begin{cases} 1/5 & a_1 = a_2 = \{f_2\} \\ 1/5 & a_1 = \{f_1\}, a_2 = \{f_2\} \text{ or } a_1 = \{f_2\}, a_2 = \{f_1\} \\ 1/5 & a_1 = \{f_1, f_2\}, a_2 = \{f_1\} \text{ or } a_2 = \{f_1, f_2\}, a_1 = \{f_1\} \\ 0 & \text{otherwise} \end{cases}$$

The reward information we can receive from the dataset in the agent-level feedback setting includes: $R^{f_2}(2), R^{f_1}(1), R^{f_1}(1), R^{f_1}(2) + R^{f_2}(1)$ and $R^{f_1}(2)$. Hence we can compute the NE directly. In the game-level feedback setting, we only know $R^{f_2}(2), R^{f_1}(1) + R^{f_2}(1)$ and $2R^{f_1}(2) + R^{f_1}(1)$. Hence **ALG** cannot distinguish these two games. Suppose the output of **ALG** selects action that 2 players select $f_1$ with probability $p$, then it is at least $(1-p)/2$-approximate NE for game 5 and at least $p/2$-approximate NE for game 6. In conclusion, **ALG** is at least $1/4$-approximate NE strategy no matter how much data is collected. $\qquad\square$

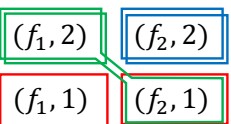

Figure 4: Facility coverage condition for $\rho$. Similar to Figure 3.

### D.1 OMITTED CALCULATIONS IN SECTION 5

**Lemma 6.** *If $n \geq 8\log((3F+1)/\delta)(3F+1)$, with probability $1-\delta$, $N(\boldsymbol{a}) \geq \rho(\boldsymbol{a})n/2 = n/2(3F+1)$ for all $\boldsymbol{a}$ with $\rho(\boldsymbol{a}) > 0$.*

The proof is identical to Lemma 4 except that we have at most $3F+1$ actions to cover instead of at most $mF+1$ actions.

**Proposition 5.** *If $n \geq F$, the maximum value of $C_{overall}$ is no smaller than $1/2F$.*

*Proof.* The proof is very similar to that of Proposition 3 except that here we have $A(\boldsymbol{a})$ instead of $A_i(\boldsymbol{a})$ $\qquad\square$

**Proposition 6.** *If $n \geq 8\log((3F+1)/\delta)(3F+1)$ and $C_{game} = 1/24F^3$, then with probability $1-\delta$, Assumption 5 holds for the exampled described in Remark 2.*

*Proof.* The procedure is similar to that in the proof of Proposition 4. Let us focus on some specific pure strategy $\pi$ choosing $\tilde{a}$ deterministically and player $i$. To calculate the reward from one facility, we need two actions. Suppose $\tilde{a}_i$ covers $f_1, f_2, \cdots, f_s$ with configuration $n_1, n_2, \cdots, n_s$. Let the action vector corresponding to facility $f_j$ be $A_{j,1}$ and $A_{j,2}$. Without loss of generality, suppose the reward from the individual facility is $\langle A_{j,1} - A_{j,2}, \theta \rangle / n_j$. Then we can write

$$A_i(\tilde{\boldsymbol{a}}) = \sum_{j \in [s]} \frac{A_{j,1} - A_{j,2}}{n_j}$$

It suffices to show

$$\frac{n}{2F(3F+1)} \sum_{j \in [s]} \left( A_{j,1}A_{j,1}^\top + A_{j,2}A_{j,2}^\top \right) \succeq C_{game} n A_i(\tilde{\boldsymbol{a}}) A_i(\tilde{\boldsymbol{a}})^\top.$$

Note that because $\{A_{j,1}, A_{j,2}\}$ may have repeated elements and repeats at most $F$ times, so we further discount the number of samples on the left hand side by $F$. Following similar procedure in the proof of Proposition 6 and $n_j \geq 1, s \leq F$ we get it suffices to show

$$\frac{n}{2F(3F+1)} \geq C_{game} n 2F$$

$\qquad\square$

## E EXPERIMENT

We implement algorithm 1 for the facility-level feedback setting and test its performance on a didactic example to verify our theory. In this section, we aim to answer three questions: (i) Does our algorithm recover the NE from datasets that satisfy assumption 3? (ii) How fast does our algorithm converge? (iii) How does the convergence rate vary with the number of players $m$?

### E.1 SETTING

The Braess paradox (Braess et al., 2005) is a famous example of congestion game. The game presented here is a modified version but retains the core idea of the paradox. There are $m$ cars wanting to travel simultaneously from S to D. There are five roads (facilities) on the road map, indexed from 0 to 4 as illustrated in figure 5. It takes some time to travel from the starting point S to the destination point D and everyone wants to accomplish it as quickly as possible. The reward (negative latency) for each facility is as follows.

$$r^0 \sim -\frac{n^1(\boldsymbol{a})}{m+1} + \eta_0, r^1 \sim -1 + \eta_1, r^2 \sim \eta_2, r^3 \sim -1 + \eta_3, r_4 \sim \frac{n^4(\boldsymbol{a})}{m+1} + \eta_4$$

where $\eta_0, \eta_1, \eta_2, \eta_3, \eta_4$ are i.i.d. random variables with Gaussian distribution of mean 0 and variance 1. Formally, the facility set and the action set of this game are $\mathcal{F} = \{0, 1, 2, 3, 4\}, \mathcal{A} = \{\{0,1\}, \{0,2,4\}, \{3,4\}\}^m$. The NE for this game is $\{0,2,4\}^m$. An interesting fact about the game

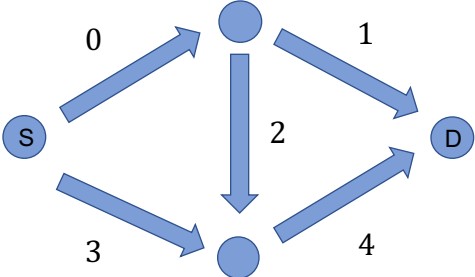

Figure 5: the Braess Paradox

is that when $m > 2$ if we remove facility 2, the road that does provide zero latency on average, the NE of this game gives everybody less latency. This means constructing more roads may aggravate traffic jams if we assume all drivers are selfish. In this paper, we employ the original game to test our algorithm.

We use two different exploration policies to collect datasets. The first policy is random exploration, where each player uniformly randomly selects his/her action from $\mathcal{A}_i$. Formally for any player $i$

$$\pi_i^{\text{random}}(a_i) = \begin{cases} 1/3 & a_i = \{0,1\} \\ 1/3 & a_i = \{0,2,4\} \\ 1/3 & a_i = \{3,4\} \end{cases}$$

This policy covers all possible facility configurations and $C_{\text{facility}} = 3^m$. The second policy is as follows.

$$\pi_0^{\text{facility}}(a_0) = \begin{cases} 1/3 & a_0 = \{0,1\} \\ 1/3 & a_0 = \{0,2,4\} \\ 1/3 & a_0 = \{3,4\} \end{cases}$$

$$\pi_i^{\text{facility}}(a_i) = \mathbb{1}\{a_i = \{0,2,4\}\} \quad \text{for } i \neq 0$$

For $m > 1$, this policy does not satisfy assumption 1 as it only covers the actions of NE unilaterally deviated by player 1. However, it covers all facility configurations achievable from unilaterally deviated actions, hence satisfying assumption 2. Moreover, since the NE covered is a pure strategy, it satisfies assumption 6 and $C_{\text{facility}}^{\text{pure}} = C_{\text{facility}} = 3$.

### E.2 RESULT

For $m = 1, 2, 3, 4, 5, 6$ and two exploration policies, we test the algorithm for different sample sizes $n$ and evaluate the performance gap of the output policy. Results are visualized in Figure 6 and 7. For each $n$, we run the algorithm on 16 datasets sampled from different random seeds. Mean performance gaps are shown with solid lines in the figures. We set $\delta = 10^{-2}$.

When the dataset is collected by $\pi^{\text{random}}$, the algorithm can always find the optimal policy as long as the dataset is large enough. Fixing $m$, as $n$ increases, the gap drops quickly when $n$ is small and gets more slowly when $n$ becomes large. This complies with the $1/\sqrt{n}$ term in the bound from theorem 3. As $m$ increases, convergence becomes slower, which also complies with theorem 3.

When the dataset is collected by $\pi^{\text{facility}}$, the algorithm can always find the optimal policy as well. Furthermore, the size of dataset needed is far smaller than that for $\pi^{\text{random}}$. Except for the $m = 1$ case the convergence rate barely varies with $m$, which complies with theorem 8.

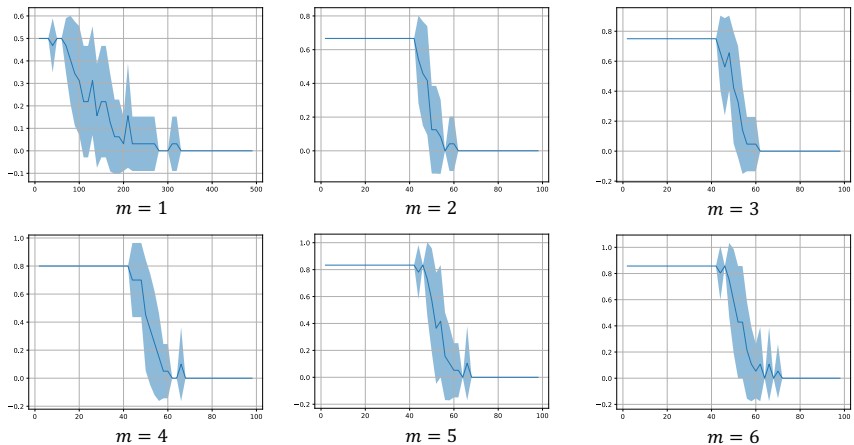

Figure 6: Result for $\pi^{\text{facility}}$. The horizontal axis is sample size and the vertical axis is the performance gap. Solid lines are means of gap over 16 seeds. Upper bounds and lower bounds of the light blue regions represent the standard deviation.

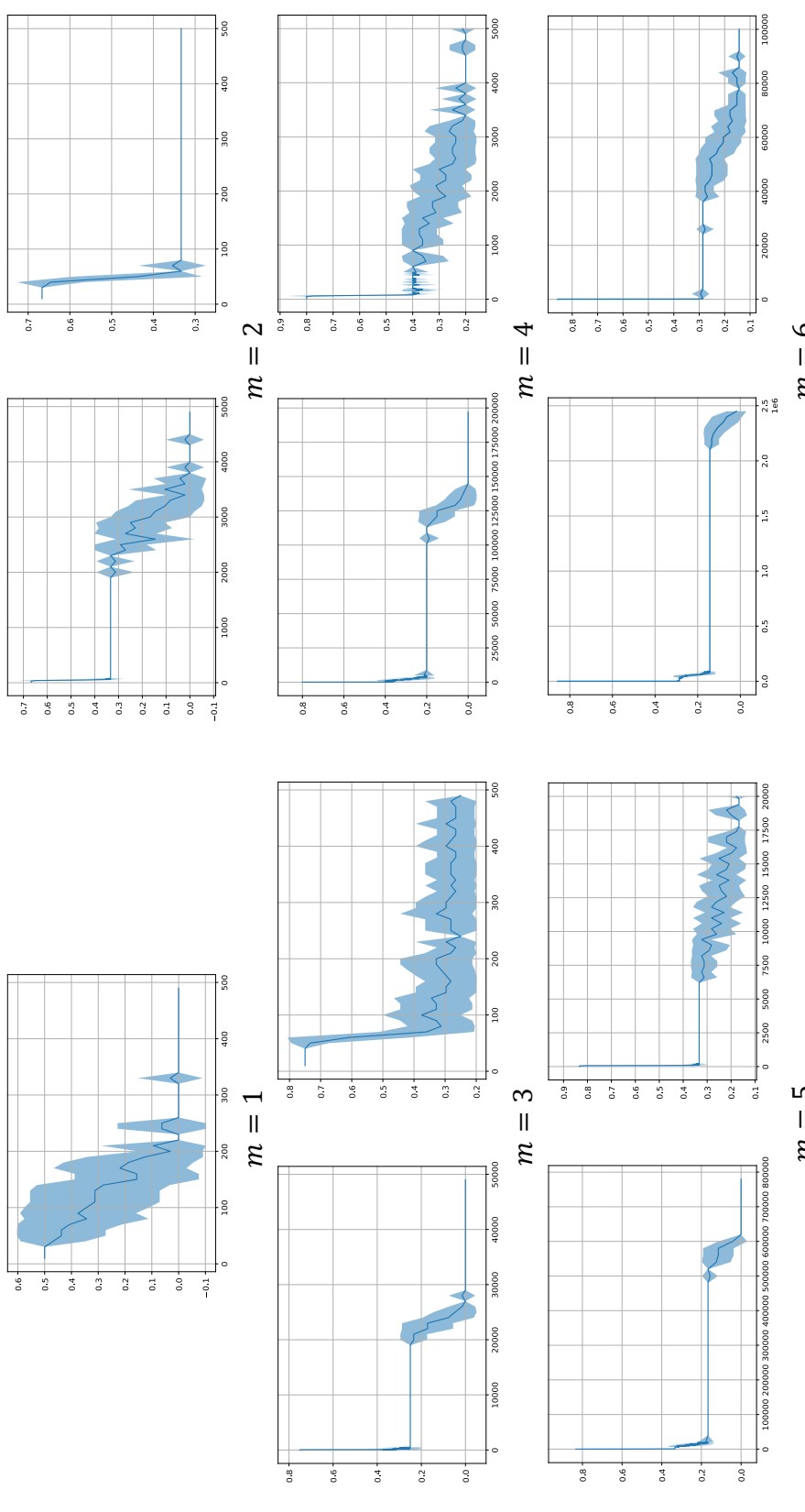

Figure 7: Result for $\pi^{\mathrm{random}}$ . For $m > 2$, we provide two figures. The global view is on the left which shows the algorithm converges to a NE policy. The local view is on the right to show the curve for small $n$ in detail.

