# OpenReview forum: "Offline Congestion Games: How Feedback Type Affects Data Coverage Requirement"
_ICLR.cc/2023/Conference — ICLR 2023 poster_

### Official Review · Reviewer_oumy · 2022-10-24

**Confidence:** 3
**Correctness:** 4
**Technical Novelty And Significance:** 3
**Empirical Novelty And Significance:** Not applicable
**Recommendation:** 8

**Clarity, Quality, Novelty And Reproducibility:**

The paper is well-organized and easy to follow. The data coverage condition needed to learn NE for congestion games is quite novel.

**Strength And Weaknesses:**

Strength:
1. Learning NE from offline samples is a rising direction and the curse of large action set would be a general issue if applying the idea to specific games. The paper provides clear discussion on congestion games.
2. The theoretical contributions are concrete.

Weaknesses:
1. The analysis on different types of feedback relies on much knowledge about the congestion game. Basically, these types are strongly related to congestion games. Thus, the techniques may not be able to inspire works on other games.

**Summary Of The Paper:**

The paper studies the conditions for recovering approximate NE of offline congestion games from samples under different types of feedback. Leveraging the structures of congestion games, new data converage assumptions are proposed to prove NE learnability, overcoming the exponentially growth of the sample complexity if directly applying the condition for general general-sum games.

**Summary Of The Review:**

The paper studies an important problem, studing the data coverage condition required for leanring NE from offline samples for congestion games with different types of feedback. Although maybe restricted to the special class of games, the theoretical analysis is concrete. Overall, the paper is of high quality and I would recommend accecption.

---

> ### Author Response · Authors · 2022-11-12
> **Reply to Reviewer oumy**
>
> Thank you very much for your careful review and appreciation! We agree that the results and techniques in our paper are specialized to congestion games and hence able to exploit the structure specific to congestion games, but we hope our work can shed new light on other types of games.

---

### Official Review · Reviewer_g7gc · 2022-10-25

**Confidence:** 4
**Correctness:** 4
**Technical Novelty And Significance:** 3
**Empirical Novelty And Significance:** 3
**Recommendation:** 8

**Clarity, Quality, Novelty And Reproducibility:**

The paper is very clear to follow, nicely written and the setup and results are the first of their kind.
No experiments are performed to assess reproducibility.

**Strength And Weaknesses:**

I think the authors consider a relevant problem, e.g. arising in modern routing networks, and place their work nicely into the existing literature. Moreover, the paper is nicely written and contributions are spelled out clearly.
The results are also sound, and the authors give a good intuition about the different settings and resulting guarantees.

What I did not like about the paper is:
- Depending on the feedback, the authors come up with  (often necessary) data coverage assumptions, under which the resulting sample complexities are derived. However, it is not obvious to me how easy is for such assumptions to be true in practice. More specifically, it would be nice to provide (even at an intuitive level) data collection strategies that will satisfy such assumptions. Unless I am missing some details, I think this would be a very nice addition to the paper.
- No experiments are performed to illustrate the developed technique and how such bounds translate into practice.

**Summary Of The Paper:**

The paper studies the problem of computing an approximate Nash equilibrium in congestion games from offline game data.
The authors consider three different types of data, depending on whether rewards are observed at the facility level, player level, or game level, and analyze their corresponding hardness in terms of required number of samples and data coverage assumptions. Moreover, to compute such approximate NE, they adapt the surrogate minimization approach of Cui and Du (2022) and decline it for the three different settings.

**Summary Of The Review:**

Overall, I think the paper is quite original and could be interesting to the algorithmic game theory community.
Moreover, I have also appreciated the counter examples designed by the authors to show separability of the different feedback models and provide intuition. Thus, I believe it deserves acceptance.
However, I would like the authors to consider the weaknesses outlined above and I am happy to increase my score upon their response.

---

> ### Author Response · Authors · 2022-11-12
> **Reply to Reviewer g7gc**
>
> Thank you very much for your careful review and appreciation!
> - **Assumptions:** From the theoretical perspective, there always exists some exploration policy such that the assumptions mentioned are satisfied with a small coverage coefficient. For instance, in Proposition 2 we show that $C_{\text{facility}}$ can be as small as $3$ for any congestion game. From the practical perspective, we agree that it is hard to guarantee that the exploration policy satisfies these assumptions with a small coefficient and it would be an interesting problem to consider how we can collect a dataset satisfying these assumptions. One promising setting we are considering is that all the players are using some no-regret algorithm, which is natural from the game theory perspective. For some specific no-regret algorithms, the joint strategy of the players will converge to a Nash equilibrium in congestion games with agent-level feedback (Heliou et al., 2017;Krichene et al., 2015). As we can learn the Nash equilibrium by running these online no-regret algorithms, we should be able to learn the Nash equilibrium by leveraging the dataset generated by these algorithms. We believe proving this conjecture can be an important contribution to offline matrix games.

---

> > ### Comment · Reviewer_g7gc · 2022-11-24
> > **Acknowledgement**
> >
> > Thank you for your response. Indeed, it would be interesting to prove the last conjecture in connection with your results.
> > I have also appreciated the additional experiments and thus I will increase my score.

---

> > > ### Author Response · Authors · 2022-11-28
> > > **Thank you**
> > >
> > > Thank you very much for your appreciation!

---

### Official Review · Reviewer_Kv1e · 2022-10-26

**Confidence:** 2
**Correctness:** 3
**Technical Novelty And Significance:** 3
**Empirical Novelty And Significance:** 4
**Recommendation:** 8

**Clarity, Quality, Novelty And Reproducibility:**

The paper is well-written and of good quality. The three types of feedback setups studied in the paper are new.

**Strength And Weaknesses:**

Strength:

The paper presents a rich collection of results for the problem of offline Nash equilibrium learning in congestion games. The paper is well-written and easy to follow.

Weakness:


Minor issues:

In page 4 when you write $R^f(\cdot | n ) \in [-1,1]$ is a bit confusing since $R^f$ is a distribution. It might be better to write the support of the distribution $R^f$ is [-1,1].

Assumptions 3 and 4 are justified only for specific examples. Can we provide conditions on the exploration policy so that they hold with high probability?


**Summary Of The Paper:**

The paper studies offline learning of the Nash equilibrium of the congestion game. In a congestion game, agents choose which facility to use. The reward the agent obtains from using the facility depends only on the number of total agents using that facility. In this paper, the authors consider three types of feedback: agent, facility, and game-level rewards. The major technical tool used in the paper is a meta-result (Theorem 1 in the paper) from Cui & Du (2022b), which reduces the problem to the one of designing a bonus function. The author proposes several coverage conditions on the exploration policy so that the meta algorithm from Cui & Du (2022b) learns the Nash equilibrium of the congestion game. Some coverage conditions are complemented with impossibility results to justify the imposing of such conditions.

**Summary Of The Review:**

I suggest an accept. However, I'm not expert in congestion game so I chose a confidence score of 2.

---

> ### Author Response · Authors · 2022-11-12
> **Reply to Reviewer Kv1e**
>
> Thank you very much for your careful review and appreciation! We have modified the definition of the reward distribution to make it more rigorous.
> - **Assumptions:** From the theoretical perspective, there always exists some exploration policy such that the assumptions mentioned are satisfied with a small coverage coefficient. For instance, in Proposition 2 we show that $C_{\text{facility}}$ can be as small as $3$ for any congestion game. From the practical perspective, we agree that it is hard to guarantee that the exploration policy satisfies these assumptions with a small coefficient and it would be an interesting problem to consider how we can collect a dataset satisfying these assumptions. One promising setting we are considering is that all the players are using some no-regret algorithm, which is natural from the game theory perspective. For some specific no-regret algorithms, the joint strategy of the players will converge to a Nash equilibrium in congestion games with agent-level feedback (Heliou et al., 2017;Krichene et al., 2015). As we can learn the Nash equilibrium by running these online no-regret algorithms, we should be able to learn the Nash equilibrium by leveraging the dataset generated by these algorithms. We believe proving this conjecture can be an important contribution to offline matrix games.

---

### Official Review · Reviewer_3Ebi · 2022-11-01

**Confidence:** 4
**Correctness:** 3
**Technical Novelty And Significance:** 3
**Empirical Novelty And Significance:** Not applicable
**Recommendation:** 6

**Clarity, Quality, Novelty And Reproducibility:**

As I mentioned above, the paper is novel and has great reproducibility, but it lacks clarity in some parts.

As a side note, I have identified some typos, a handful of which I enumerate below at the authors' convenience. However, I believe a careful pass is required.

Page 6: selects -> selects it
Page 6: with -> where
Page 14: an joint action -> a joint action

**Strength And Weaknesses:**


Strengths:

1) The authors take recent work's particularly by Ciu and Du on finding a minimal dataset coverage assumption to learn an NE in a general-sum matrix to the realm of congestion games.
2) The results are novel to the best of my knowledge.
3) Given the time constraints, I verified a large part of the mathematical work, and their work is mathematically sound to the best of my knowledge.


Weaknesses:

1) The impossibility results (Theorem 4 and 6) require the assumption of agents whose allowable sets of actions includes the empty set. This is a highly non-standard assumption in congestion games/game theory. Is this assumption necessary for these proofs or is there a workaround?
2) Various parts of the paper need more clarification or/and notational precision. For example, what is $n$ in Assumption 2? Do we assume that the condition holds for some $n$ or all possible $n$? Of course, the following paragraph clarifies the issue, but the assumption is not self-explanatory.
3) The lack of experimental results is disappointing. Although this work is theoretical, it should be relatively easy to add experimental results showing how close the theoretical bound is to the actual gap.
4) Even in the toy examples provided, adding the actual scale of the required data would be very beneficial, even if just for clarity. For example, in Figure 1, what is the actual number of data required to determine this equilibrium?



**Summary Of The Paper:**

In their work, the authors focus on congestion games and examine information theoretic conditions for identifying NE. Specifically, they identify the minimal assumptions required to efficiently determine if a policy is a Nash Equilibrium of a congestion game from a given dataset. The authors examine 3 different models, one with access to the costs of individual facilities, one with access to agent level costs, and one with access to the sum of agents' costs. The required assumptions depend on the information encompassed in the dataset, and stronger assumptions are naturally required when less information is provided.


**Summary Of The Review:**

A theoretical and mathematically sound paper, which lacks clarification in some parts but requires further work regarding experimental results and given examples. I would be happier if the impossibility results did not require the assumption of agents with having the choice of an empty strategy.

---

> ### Author Response · Authors · 2022-11-12
> **Reply to Reviewer 3Ebi**
>
> Thank you very much for your careful review and constructive suggestions! Please find our response to your questions and concerns below:
> - **Empty set action in impossibility results:** In the revised paper, we construct new hard instances which do not depend on empty-set action for both Theorem 4 and 6. The intuition is to construct hard instances such that they are distinguishable with stronger feedback but indistinguishable with weaker feedback.
> - **Clarification of Assumption 2:** Sorry for the ambiguity in the assumption. In Assumption 2, $n$ is the configuration of the facility, i.e., the number of players choosing the facility. Assumption 2 holds for all possible $n$. We have checked and revised the whole paper to clarify those details. Please let us know if there is still any ambiguity or impreciseness.
> - **Data scale in toy examples:** We provide several examples (Figure 1&2, Remark 1&2) in the paper to better illustrate the proposed assumptions. We do not intend to give the sample complexity as it would be rather complicated to define the whole game and the sample complexity will be just a direct application of the theorems in the paper. Instead, these examples only aim to give some intuition about the assumptions.

---

> > ### Comment · Reviewer_3Ebi · 2022-11-24
> > **Response**
> >
> > On page 14 proof of Theorem 2 you write " Consider congestion game with a single facility f and five players. The action space for each
> > player is {∅, {f}}. So it seems that you are still using the emptyset as an option?
> >
> > Also, it is necessary for your construction to have an agent have in their strategysets sets of actions such that one of them is a strict subset of another?
> >
> > Although this is less "unnatural" than having the emptyset as an action it is still very much non-standard in all examples of congestion games that I can think of. E.g. in standard cost driven congestion games any such action that is a strict superset of another would be strictly dominated by its subset hence it does not make for a meaningful option. So, how critical is this gadget as well the possibility of having both positive and negative utilities?
> >
> > Thank you again for your responses!

---

> > > ### Author Response · Authors · 2022-11-28
> > > **Reply to Reviewer 3Ebi (Part I)**
> > >
> > > Thank you very much for scrupulously reading our proof and providing constructive suggestions. Please find our response below:
> > >
> > > First, we would like to point out that we intend to analyze congestion games in the most general setting. Hence we did not impose any constraint on action set or positivity of utility functions. Having utility functions with both positive and negative values  in the counter-examples, it is straightforward to demonstrate where the separations among different feedback types come from and why Assumption 2 cannot be strictly relaxed.
> > >
> > > Second, although games with negative utility functions and without empty actions arise often, it is also possible that these two constraints are broken in some real-world applications. Here we provide a concrete example. There is a platform that provides commercial positions on several websites (facilities). Several companies (agents) choose websites to advertise their products. Some money is charged if a position is taken. Since advertising promotes sales, taking a position can yield positive utility. However, when too many commercials appear on the same website, this promotion becomes weak and may fall below the fee to pay for the position, yielding a negative utility. A company can choose not to advertise its product since the platform is too congested, which corresponds to an empty action.
> > >
> > > Third, we construct three pairs of games that can replace the counter-examples that appear in the proof of Theorem 2, 4 and 6. In these games no action is a strict subset of another action and all utilities are negative, which we believe is natural even in cost-driven congestion games.
> > >
> > > - *Theorem 2*: Consider games with three facilities and two players. The action set is $\mathcal A=\lbrace\lbrace f_1,f_2\rbrace,\lbrace f_1,f_3\rbrace,\lbrace f_2,f_3\rbrace\rbrace$. The two games are constructed as follows. NEs are marked in bold font.
> > >
> > >   | $R^{f_1}(2)=-0.45$              | $\boldsymbol{R^{f_2}(2)=-0.05}$ | $R^{f_3}(2)=-0.4$              |
> > >   | ------------------------------- | ------------------------------- | ------------------------------ |
> > >   | $\boldsymbol{R^{f_1}(1)=-0.35}$ | $R^{f_2}(1)=-0.1$               | $\boldsymbol{R^{f_3}(1)=-0.2}$ |
> > >
> > >   | $R^{f_1}(2)=-0.45$ | $\boldsymbol{R^{f_2}(2)=-0.05}$ | $\boldsymbol{R^{f_3}(2)=-0.05}$ |
> > >   | ------------------ | ------------------------------- | --------------------------- |
> > >   | $R^{f_1}(1)=-0.35$ | $R^{f_2}(1)=-0.1$               | $R^{f_3}(1)=-0.2$           |
> > >
> > >   The exploration policy $\rho$ covers all configurations but $f_3$ with two players selecting it. In this case these two games are indistinguishable but have different NEs.
> > > - *Theorem 4*: Consider games with three facilities and two players. The action set is $\mathcal A=\lbrace\lbrace f_1,f_2\rbrace,\lbrace f_1,f_3\rbrace,\lbrace f_2,f_3\rbrace\rbrace$. The two games are constructed as follows. NEs are marked in bold font.
> > >
> > >   | $R^{f_1}(2)=-0.3$              | $\boldsymbol{R^{f_2}(2)=-0.2}$ | $R^{f_3}(2)=-0.9$              |
> > >   | ------------------------------- | ------------------------------- | ------------------------------ |
> > >   | $\boldsymbol{R^{f_1}(1)=-0.6}$ | $R^{f_2}(1)=-0.05$               | $\boldsymbol{R^{f_3}(1)=-0.2}$ |
> > >
> > >   | $\boldsymbol{R^{f_1}(2)=-0.2}$ | $\boldsymbol{R^{f_2}(2)=-0.1}$ | $R^{f_3}(2)=-0.8$ |
> > >   | ------------------ | ------------------------------- | --------------------------- |
> > >   | $R^{f_1}(1)=-0.7$ | $R^{f_2}(1)=-0.15$               | $R^{f_3}(1)=-0.3$           |
> > >
> > >   The exploration policy $\rho$ covers $(a_1,a_2)=(\lbrace f_1,f_2\rbrace,\lbrace f_1,f_3\rbrace),(\lbrace f_1,f_2\rbrace,\lbrace f_2,f_3\rbrace),(\lbrace f_1,f_3\rbrace,\lbrace f_2,f_3\rbrace)$. All configurations are covered, so it is easy for dataset with facility-level feedback to recover the whole game. However all agent-level feedbacks are identical for these two games. Specifically, $R^{f_1}(2)+R^{f_2}(1)=-0.35,R^{f_1}(2)+R^{f_3}(1)=-0.5,$$R^{f_1}(1)+R^{f_2}(2)=-0.8,R^{f_2}(2)+R^{f_3}(1)=-0.4,$$R^{f_1}(1)+R^{f_3}(2)=-1.5,R^{f_2}(1)+R^{f_3}(2)=-0.95$.

---

> > > > ### Author Response · Authors · 2022-11-28
> > > > **Reply to Reviewer 3Ebi (Part II)**
> > > >
> > > > - *Theorem 6*: Consider games with four facilities and two players. The action set is $\mathcal A=\lbrace\lbrace f_1,f_2,f_3\rbrace,\lbrace f_1,f_2,f_4\rbrace,\lbrace f_1,f_3,f_4\rbrace,\lbrace f_2,f_3,f_4\rbrace\rbrace$. The two games are constructed as follows. NEs are marked in bold font.
> > > >
> > > >   | $R^{f_1}(2)=-0.7$             | $\boldsymbol{R^{f_2}(2)=-0.5}$ | $\boldsymbol{R^{f_3}(2)=-0.15}$ | $R^{f_4}(2)=-0.6$                 |
> > > >   | ------------------------------ | ------------------------------ | ------------------------------- | ------------------------------- |
> > > >   | $\boldsymbol{R^{f_1}(1)=-0.25}$ | $R^{f_2}(1)=-0.2$              | $R^{f_3}(1)=-0.1$              | $\boldsymbol{R^{f_4}(1)=-0.6}$ |
> > > >
> > > >   | $\boldsymbol{R^{f_1}(2)=-0.45}$ | $R^{f_2}(2)=-0.5$             | $\boldsymbol{R^{f_3}(2)=-0.4}$ | $\boldsymbol{R^{f_4}(2)=-0.35}$                 |
> > > >   | ------------------------------ | ------------------------------ | ------------------------------- | ------------------------------- |
> > > >   | $R^{f_1}(1)=-0.25$              | $R^{f_2}(1)=-0.7$ | $R^{f_3}(1)=-0.6$              | $R^{f_4}(1)=-0.1$ |
> > > >
> > > >   The exploration policy $\rho$ covers $(a_1,a_2)=(\lbrace f_1,f_2,f_3\rbrace,\lbrace f_2,f_3,f_4\rbrace),(\lbrace f_1,f_2,f_4\rbrace,\lbrace f_1,f_3,f_4\rbrace),(\lbrace f_1,f_2,f_3\rbrace,\lbrace f_1,f_3,f_4\rbrace),(\lbrace f_1,f_2,f_3\rbrace,\lbrace f_1,f_2,f_3\rbrace),(\lbrace f_2,f_3,f_4\rbrace,\lbrace f_2,f_3,f_4\rbrace)$. It can be verified that by having agent-level feedback, we can solve a well-posed linear equation system to recover the whole game. However all game-level feedbacks are identical for these two games. Specifically, $R^{f_1}(1)+2R^{f_2}(2)+2R^{f_3}(2)+R^{f_4}(1)=-2.15,$$2R^{f_1}(2)+R^{f_2}(1)+R^{f_3}(1)+2R^{f_4}(2)=-2.9,$$2R^{f_1}(2)+R^{f_2}(1)+2R^{f_3}(2)+R^{f_4}(1)=-2.5,$$2R^{f_1}(2)+2R^{f_2}(2)+2R^{f_3}(2)=-2.7,$$2R^{f_2}(2)+2R^{f_3}(2)+2R^{f_4}(2)=-2.5$.
> > > >
> > > > We will incorporate these new results in the final version of our paper. If you have any other questions/suggestions to the counter-examples or other parts of the paper, please let us know!

---

> > > > > ### Comment · Reviewer_3Ebi · 2022-12-07
> > > > > **Response**
> > > > >
> > > > > Thank you for the updated response. This example is more satisfactory indeed and I will update my score accordingly.

---

### Author Response · Authors · 2022-11-12
**Paper Revision**

We have revised the paper according to the suggestions of the reviewers. Specifically, we addressed all the minor issues, clarified Assumption 2, and modified the proof of Theorem 4 and Theorem 6 so that empty set action is not utilized. Please let us know if there are any further suggestions.

---

### Author Response · Authors · 2022-11-19
**Additional Numerical Experiments**

We added a new section for experiments to the paper (Section D in the Appendix). In our experiments, we implement our algorithm for the facility-level feedback setting and test it on a well-known congestion game (Braess et al., 2005). We test our algorithm with two kinds of datasets. The first one is collected by uniformly random exploration, which is common in practice. The other one is specially designed such that it satisfies the newly proposed Assumption 2 but does not satisfy Assumption 1. This dataset is used to verify the learnablility under the new assumption. The results are briefly summarized as follows:
- With dataset from random exploration, the algorithm always manages to find the NE. When the number of players increases, the amount of data required to find the NE also increases.
- With the other deliberately designed dataset, the algorithm finds the NE as well, verifying the learnability under Assumption 2. However, when the number of players increases, the amount of data needed to find the NE barely changes. This phenomenon is explained in the newly added Theorem 8 (in section A.1 of the appendix).

In summary, the experiments conducted verify the correctness of our algorithm under newly proposed assumption. Phenomena arised in experiments are also well-explained by our theory.

---

### Decision · Program_Chairs · 2023-01-20

**Decision:**

Accept: poster

**Justification For Why Not Higher Score:**

Results are a bit restricted

**Justification For Why Not Lower Score:**

The contributions are valuable and new

**Metareview: Summary, Strengths And Weaknesses:**

The paper looks at approximating NE of offline congestion games from samples under feedback of bandits type (and other).

The paper is well written, and very well motivated. The major criticism is that the knowledge assumptions might appear a bit too strong, but it is still nonetheless non-triviial and worth reading

**Note From Pc:**

if the above contains the word "oral" or "spotlight" please see: "oral" presentation means -> notable-top-5% and "spotlight" means -> notable-top-25%. As stated in our emails, we are disassociating presentation type from AC recommendations